# Reinforcement Learning with Random Time Horizons

**Enric Ribera Borrell** [* 1 2]  **Lorenz Richter** [* 1 3]  **Christof Schütte** [1 2]

## Abstract

We extend the standard reinforcement learning framework to random time horizons. While the classical setting typically assumes finite and deterministic or infinite runtimes of trajectories, we argue that multiple real-world applications naturally exhibit random (potentially trajectory-dependent) stopping times. Since those stopping times typically depend on the policy, their randomness has an effect on policy gradient formulas, which we (mostly for the first time) derive rigorously in this work both for stochastic and deterministic policies. We present two complementary perspectives, trajectory or state-space based, and establish connections to optimal control theory. Our numerical experiments demonstrate that using the proposed formulas can significantly improve optimization convergence compared to traditional approaches.

## 1. Introduction

The goal in reinforcement learning is to identify policies $\pi$ that maximize the expected return

$$J(\pi) = \mathbb{E}_\pi \left[ \sum_{n=0}^{N} r_n(S_n, A_n) \right], \qquad (1)$$

where the states $S_n$ and the actions $A_n$ run until a time $N$. While the classical framework considers the case where $N$ is either finite and deterministic or infinite, runtimes can as well be random. To name two examples, one could consider robots that aim to reach a target in a noisy environment or one could play a game whose terminating state depends on reaching a certain level. Furthermore, considering stochastic policies (as commonly done in practice in order to balance exploration and exploitation) leads to the fact that runtimes

are random even in deterministic environments. In all those cases the termination depends on previous (random) choices and is therefore random itself. However, the common literature has so far largely ignored the fact that runtimes can be random. In particular, policy gradient theorems typically assume that they are either deterministic or infinite (Sutton & Barto, 2018). Nonetheless, it is intuitively evident that considering random times should have an effect on optimization formulas since the runtimes mostly depend on the current policy such that in principle one should be careful with taking gradients (cf. Nota & Thomas (2020)).

In this work, we close this theoretical gap and systematically investigate the effect of random time horizons in reinforcement learning problems. Our contributions can be summarized as follows:

- We extend the typical reinforcement learning framework to random time horizons, including (deterministic) finite and infinite runtimes as special cases.

- We state corresponding (random time) policy gradient theorems both for deterministic and stochastic policies.

- This allows us for the first time to rigorously derive a trajectory-based gradient estimator for random time horizons (for stochastic as well as deterministic policies) in discrete time.

- For deterministic policies we derive a novel model-based policy gradient formula, which does not rely on learning the $Q$-value function as in actor-critic approaches.

- In multiple numerical experiments, we systematically investigate the effect of incorporating the randomness of the time horizon in the gradient computation. For most cases we can see significant performance improvements of our gradient formulas compared to the standard ones, in particular in terms of convergence speed.

### 1.1. Related work

Gradient based on-policy optimization in reinforcement learning has been pioneered by Sutton et al. (1999), where a policy gradient theorem for infinite time horizons and

---

*Equal contribution [1]Zuse Institute Berlin, 14195 Berlin, Germany [2]Institute of Mathematics, Free University Berlin, 14195 Berlin, Germany [3]dida Datenschmiede GmbH, 10827 Berlin, Germany. Correspondence to: Enric Ribera Borrell <ribera.borrell@zib.de>, Lorenz Richter <richter@zib.de>.

*Proceedings of the 42$^{nd}$ International Conference on Machine Learning*, Vancouver, Canada. PMLR 267, 2025. Copyright 2025 by the author(s).

discrete state spaces has been suggested. In Silver et al. (2014) it has been extended to deterministic policies and continuous state-spaces, still in infinite time. In the seminal monograph from Sutton & Barto (2018) random runtimes are mentioned, however, not studied. To the best of our knowledge, the only work that studies trajectory-dependent random runtimes is from Bojun (2020). However, it solely focuses on discrete state and action spaces and considers only stochastic policies, mostly relying on Markov chain theory for proving the statements. Our work, in contrast, operates in continuous spaces and adds formulas for deterministic policies. For further work on random time horizons in reinforcement learning, we refer to Mandal et al. (2023) and Chen et al. (2024), who, however, only study trajectory-independent stopping times.

We further refer to Nota & Thomas (2020), who show that taking heuristic versions of policy gradients (e.g. dropping discount factors without motivation) might lead to wrong expressions and that in general one should be careful with employing algorithms that are not backed by theory. Also, we refer to White (2017), who unifies finite and infinite time horizons by introducing generalized (transition-based) discount factors, which might in principle be enhanced to random time problems.

Finally, we want to mention that random time horizons are more popular in stochastic optimal control settings, which consider deterministic policies and are typically stated in continuous time (Pham, 2009). We want to highlight Ribera Borrell et al. (2024), who derive trajectory-based gradient estimators for random time optimal control problems. In fact, for discrete times, we can generalize their result to non-Gaussian transition densities. We refer to Lie (2021), who analyzes convexity properties of cost functionals with random stopping times, and to Zhou et al. (2021), who suggest an algorithm valid for random stopping times in optimal control that is inspired by the actor-critic method. Lastly, we refer to Schütte et al. (2023) for random stopping time problems and related optimization methods in molecular dynamics and to Quer & Ribera Borrell (2024) who approach optimal control problems appearing in molecular dynamics applications with reinforcement learning techniques.

### 1.2. Outline of the article

In Section 2 we formally state the reinforcement learning setting allowing for random runtimes. Section 2.1 comments on taking either trajectory or state-space perspectives on this setting. Using both perspectives, we then derive policy gradient theorems incorporating random time horizons in Section 2.2. In Section 3 we compare our novel gradient formulas to classical formulas in numerical experiments. Finally, Section 4 concludes and provides an outlook for future research.

## 2. Reinforcement learning with random time horizons

In the following we will formally introduce the reinforcement learning setting we consider in this article. Further details on the notation can be found in Appendix A. We consider the continuous state-space $\mathcal{S} \subset \mathbb{R}^{d_s}$ and the continuous action space $\mathcal{A} \subset \mathbb{R}^{d_a}$ and define a time-discrete *Markov decision process* $\mathcal{M} = (\mathcal{S}, \mathcal{A}, r_n, \rho_0, p)$ on the probability space $(\Omega, \mathcal{F}, \mathbb{P})$. Here the reward function $r_n : \mathcal{S} \times \mathcal{A} \to \mathbb{R}$ provides the signal that is received after being in state $S_n \in \mathcal{S}$ and having taken action $A_n \in \mathcal{A}$ at time[1] $n \in \mathbb{N}$. The function $\rho_0 : \mathcal{S} \to \mathbb{R}_{\geq 0}$ represents the initial probability density of the states and $p : \mathcal{S} \times \mathcal{S} \times \mathcal{A} \to \mathbb{R}_{\geq 0}$ is the time-homogeneous (state-action) transition probability density. To be more precise, let $\Lambda \subset \mathcal{S}$, then the probability of starting in a set $\Lambda$ is given by

$$\mathbb{P}(S_0 \in \Lambda) = \int_{\Lambda} \rho_0(s) \mathrm{d}s \qquad (2)$$

and the probability of transitioning into the set $\Lambda$ conditioned on being in state $s \in \mathcal{S}$ and having chosen action $a \in \mathcal{A}$ at time $n \in \mathbb{N}$ is given by

$$\mathbb{P}(S_{n+1} \in \Lambda \mid S_n = s, \ A_n = a) = \int_{\Lambda} p(s', s, a) \mathrm{d}s'. \quad (3)$$

A *policy* is a decision rule determining which action $A_n \in \mathcal{A}$ to take when being in state $S_n \in \mathcal{S}$ at time $n \in \mathbb{N}$. In this work, we consider stationary Markovian policies, i.e., for every time step the decision rule is the same and the choice of the action at time $n$ does not depend on the previous states $S_0, \ldots, S_{n-1}$ of the trajectory. We distinguish between two types of policies:

- A *deterministic policy* $\mu : \mathcal{S} \to \mathcal{A}$ is a function that directly maps each state to an action.

- A *stochastic policy* $\pi : \mathcal{S} \times \mathcal{A} \to \mathbb{R}_{\geq 0}$ is the probability density of taking an action in a given state. To be more precise, let $M \subset \mathcal{A}$, then the probability to choose an action in $M$ when being in state $s \in \mathcal{S}$ is given by

$$\mathbb{P}(A_n \in M \mid S_n = s) = \int_M \pi(s, a) \mathrm{d}a. \qquad (4)$$

*Remark* 2.1 (Deterministic as stochastic policy). Formally, a deterministic policy $\mu$ can be expressed as a stochastic policy via $\pi(s, a) = \delta(\mu(s) - a)$, where $\delta$ is the Dirac delta distribution. In the sequel, we will therefore often only refer to the stochastic policy $\pi$ and only mention the deterministic policy $\mu$ explicitly when necessary.

---

[1] In principle, the reward function can be explicitly time-dependent, however, for random time horizons one typically chooses it to be time-independent.

The goal in reinforcement learning is to identify policies $\pi$ that maximize the expected return (i.e. cumulative reward)

$$J(\pi) = \mathbb{E}_\pi \left[ \sum_{n=0}^{N} r_n(S_n, A_n) \right]. \tag{5}$$

We note that the randomness of the return depends on the policy $\pi$ and the densities $\rho_0, p$ defined before. For notational convenience, only the learnable $\pi$ is included in the subscript of the expectation operator (see Appendix A for more details). Typically, one either considers $N$ to be fixed and deterministic[2] or $N = \infty$. In the latter case one has to discount the reward function, e.g. via $r_n(s, a) = \gamma^n r(s, a)$, where $\gamma \in (0, 1)$ is a discount factor and $r : \mathcal{S} \times \mathcal{A} \to \mathbb{R}$ is not explicitly time-dependent. In this paper we generalize this setting to time horizons $N$ that can be random. In practice, $N$ often depends on the (random) trajectory and is defined as the first time reaching a predefined set of terminal states $\mathcal{T} \subset \mathcal{S}$, i.e.

$$N := \min\{n \in \mathbb{N} : S_n \in \mathcal{T}\}. \tag{6}$$

We note that, due to the dependency on the trajectory, $N$ also depends on the policy $\pi$. Throughout this paper, we assume that $N$ is almost surely finite, i.e. $\mathbb{P}(N < \infty) = 1$.

Interestingly, as noted already in Derman (1970) (see also Chapter 5.3 in Puterman (2014)), the discounted infinite horizon case can be considered as a random time horizon problem[3]. For convenience, we state the proof in Appendix C.

**Lemma 2.2** (Random time perspective on discounted infinite time problem). *Let $N_\gamma$ be a geometrically distributed random variable with success probability $1 - \gamma$, where $\gamma \in (0, 1)$. Then, choosing $N = N_\gamma$ and $r_n = r$ in objective* (5) *is equivalent to choosing $N = \infty$ and $r_n = \gamma^n r$.*

A notable conclusion is that infinite time horizon problems can in fact be seen as problems that are finite with probability one. To be precise, they can be interpreted as random time horizon problems where the trajectory can be arbitrarily long, however, (almost surely) not infinitely long. We note, however, that the equivalence only holds in expectation and numerical implications might have to be studied further, cf. Paternain (2018), Zhang et al. (2020) and Mandal et al. (2023).

---

[2]Markov decision processes with finite runtimes are often called *episodic*, whereas infinite time processes are called *continuing*. For the episodic case, some works include random, wheres some assume deterministic runtimes. However, computational consequences for random runtimes are typically not studied.

[3]We note, however, that due to the time-dependent reward function, the starting time matters for infinite time horizon problems in this setting.

## 2.1. Trajectory vs. state-space perspective

As commonly done, let us in the following assume the reward function to be not explicitly time-dependent, i.e. we set $r_n = r$. For the reinforcement learning problem defined above, we can consider two complementary perspectives, leading to different formulas and different implementations. We can either average the reward function (5) over trajectories, such that the expectation is taken w.r.t. *discrete path measures*, or we can view the problem on a state-space level, not inherently incorporating any dynamics. Let us elaborate in the following.

**Trajectory perspective.** This viewpoint is the one described above, i.e., we consider trajectories $(S_n, A_n)_{n=0}^{N}$ that evolve over time, given transition densities for the states and policies for the actions.

**State-space perspective.** In this viewpoint we define the state-space density $\rho^\pi$ (sometimes also called *on-policy distribution*) as the density of the process $S$ under policy $\pi$. Recall that the density is independent of time since we consider stationary problems. It can be defined as follows. First, we define the function $\rho_n^\pi$ s.t. for all $\Lambda \subset \mathcal{S}$ it holds

$$\int_\Lambda \rho_n^\pi(s)\mathrm{d}s = \mathbb{P}_\pi(S_n \in \Lambda), \tag{7}$$

so it measures the probability of being in a state at a given time[4] (to be even more precise, the above probability means $\mathbb{P}_\pi(S_n \in \Lambda) = \mathbb{P}_\pi(S_n \in \Lambda, n \leq N)$). We can now define the state-space density as the probability of being in a certain state irrespective of the time via

$$\rho^\pi := \frac{\eta^\pi}{Z^\pi}, \quad \eta^\pi := \sum_{n=0}^{\infty} \rho_n^\pi, \quad Z^\pi := \int_\mathcal{S} \eta^\pi(s)\mathrm{d}s. \tag{8}$$

In other words, $\rho^\pi$ measures the frequency of visiting a certain region at some point during the evolution of the trajectory and thus does not contain time information anymore. To make this observation more precise, the following lemma shows that the (unnormalized) function $\eta^\pi$ defined in (8) can be interpreted as counting how often trajectories stay in certain regions of the space, see Appendix C for the proof.

**Lemma 2.3.** *For a set $\Lambda \subset \mathcal{S}$ it holds*

$$\int_\Lambda \eta^\pi(s)\mathrm{d}s = \mathbb{E}_\pi \left[ \sum_{n=0}^{N} \mathbb{1}_\Lambda(S_n) \right] \tag{9}$$

*and in particular*

$$Z^\pi = \int_\mathcal{S} \eta^\pi(s)\mathrm{d}s = \mathbb{E}_\pi [N + 1]. \tag{10}$$

The state-space perspective is particularly prominent in reinforcement learning since it readily allows for off-policy

---

[4]We assume that the process $S$ stops after $N$ steps.

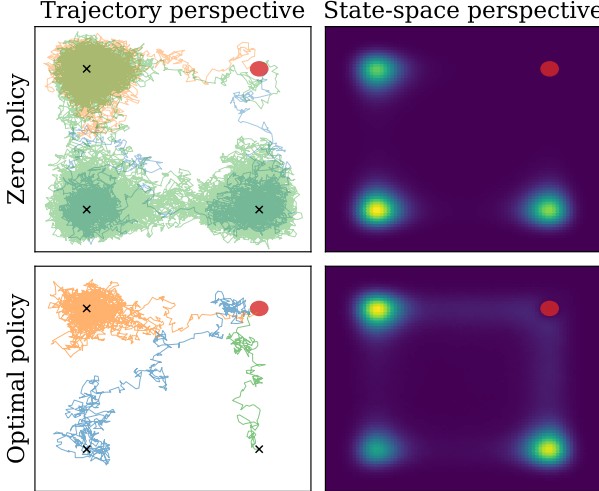

Trajectory perspective  State-space perspective

*Figure 1.* We illustrate trajectory and state-space perspectives on the reinforcement learning problem. In the left panel, we plot three trajectories that start at the black crosses, respectively, and run until hitting the target set displayed in red under the dynamics that is governed by a multi-well potential, see Section 3.3 for details. The right-hand side displays the corresponding state-space density $\rho^\pi$, as defined in (8). In the first row we choose the initial (deterministic) policy that only returns zero actions and in the second row we consider the (learned) optimal policy according to the problem defined in Section 3.3.

learning strategies, see, e.g., Degris et al. (2012); Mnih et al. (2013); Lillicrap (2016). We refer to Agarwal et al. (2022) for additional context on the state-space perspective in terms of so-called occupancy measures and refer to Figure 1 for an illustration of trajectory and state-space perspectives.

We can now state the state-space version of the expected return defined in (5), which has been derived in Theorem 4 in Bojun (2020) in a discrete space setting.

**Proposition 2.4** (State-space expected return)**.** *Assuming* $\mathbb{P}(N < \infty) = 1$*, the expected return* (5) *can be written as*

$$J(\pi) = \mathbb{E}_\pi[N+1] \, \mathbb{E}_{\substack{s \sim \rho^\pi \\ a \sim \pi(s, \cdot)}} \left[ r(s, a) \right], \qquad (11)$$

*where* $\rho^\pi$ *is the state-space density defined in* (8).

Contrary to classical results, we note the appearance of the expected runtime in (11), acting as a scaling factor. Loosely speaking, with the change from the trajectory to the state-space perspective one omits the dynamics and therefore the runtime information, and Proposition 2.4 shows the correct way of integrating this information back into the state-space perspective. We refer to Appendix C for the proof and to Remark C.1 for an alternative derivation, which might also bring some further intuition.

For infinite time horizons, often the identity

$$\bar{J}(\pi) = \mathbb{E}_{\substack{s \sim \widetilde{\eta}^\pi \\ a \sim \pi(s, \cdot)}} \left[ r(s, a) \right] \qquad (12)$$

is stated, where, in analogy to (8), $\widetilde{\eta}^\pi := \sum_{n=0}^\infty \gamma^n \rho_n^\pi$ is called *discounted state distribution*, see e.g. Silver et al. (2014). We note, however, that $\widetilde{\eta}^\pi$ is not a density since it is not normalized, and the correct expression for the infinite time case would be

$$J(\pi) = \frac{1}{1-\gamma} \mathbb{E}_{\substack{s \sim \widetilde{\rho}^\pi \\ a \sim \pi(s, \cdot)}} \left[ r(s, a) \right], \qquad (13)$$

where $\widetilde{\rho}^\pi$ is the normalized version of $\widetilde{\eta}^\pi$, thus aligning with Lemma 2.2.

*Remark* 2.5 (Sampling from $\rho^\pi$). While we call expectations w.r.t. $\rho^\pi$ state-space-based, we note that typically, in order to approximate the expectations numerically, one still needs to simulate trajectories. However, at the same time, as mentioned before, the state-space perspective allows to readily design off-policy learning strategies and to use replay buffers (cf. Degris et al. (2012)).

## 2.2. Optimization and policy gradient theorems

The notorious question in reinforcement learning is how to optimize the expected return (5) w.r.t. the policy $\pi$ (for stochastic policies) or w.r.t. the function $\mu$ (for deterministic policies). In practice, we typically assume that $\pi$ or $\mu$ are parametrized by the parameter vector $\theta \in \mathbb{R}^p$, and we may write $\pi_\theta$ or $\mu_\theta$, respectively. For ease of notation, we often omit the parameter dependence, however.

A quantity that often appears in the context of optimization is the so-called *Q-value function* defined as the expected return conditioned on a state-action pair,

$$Q^\pi(s, a) := \mathbb{E}_\pi \left[ \sum_{n=0}^N r(S_n, A_n) \middle| S_0 = s, A_0 = a \right]. \quad (14)$$

We note that starting at $n = 0$ holds without loss of generality and one could also start at any other time, since the problem is time-autonomous. Further, one can define the *value function* (or *return-to-go*) as $V^\pi(s) = \mathbb{E}_{a \sim \pi(s, \cdot)} [Q^\pi(s, a)]$. We note that for the expected return (5) it holds $J(\pi) = \mathbb{E}_{s \sim \rho_0} [V^\pi(s)]$ and refer to Appendix B for further details.

The following statement shows how one can compute gradients w.r.t. stochastic policies of expected returns that incorporate random stopping times. A proof can be found in Appendix C.

**Proposition 2.6** (Policy gradient)**.** *For the gradient of the expected return* (5) *it holds*

$$\nabla_\theta J(\pi_\theta) = \mathbb{E}_\pi \left[ \sum_{n=0}^N \nabla_\theta \log \pi_\theta(S_n, A_n) Q^\pi(S_n, A_n) \right] \tag{15}$$

$$= \mathbb{E}_\pi[N+1] \, \mathbb{E}_{\substack{s \sim \rho^\pi \\ a \sim \pi(s,\cdot)}} \left[ \nabla_\theta \log \pi_\theta(s, a) Q^\pi(s, a) \right], \tag{16}$$

*where $\rho^\pi$ is the state-space density defined in* (8).

We note that – analogous to Section 2.1 – Proposition 2.6 provides formulas following either a trajectory perspective, stated in (15), or a state-space perspective, as in (16). For the former we note that a rigorous derivation for random time horizons has to the best of our knowledge not been conducted before[5]. Further, we note that, in contrast to deterministic stopping times, one cannot simply apply automatic differentiation of the expected return w.r.t. the parameter $\theta$ for optimization due to the policy-dependency of the runtime. However, our formula (15) suggests the alternative objective

$$J_{\text{eff}}(\pi_\theta, \pi_\vartheta) := \mathbb{E}_{\pi_\vartheta} \left[ \sum_{n=0}^N \log \pi_\theta(S_n, A_n) Q^{\pi_\vartheta}(S_n, A_n) \right], \tag{17}$$

which can be (auto-)differentiated w.r.t. $\theta$, yielding $\nabla_\theta J(\pi_\theta)$ when setting $\vartheta = \theta$ after differentiation, i.e.

$$\nabla_\theta J_{\text{eff}}(\pi_\theta, \pi_\vartheta)\big|_{\vartheta=\theta} = \nabla_\theta J(\pi_\theta) \tag{18}$$

(in practice we can simply detach $S_n$ and $A_n$ from the computational graph; for the analog reasoning in the state-space perspective see Appendix D). Further, we note that the state-space-based formula (16) in fact also relies on trajectories due to the definition of the $Q$-value function in (14) – it has already been stated in Bojun (2020) in a discrete space setting. To the best of our knowledge, the state-space formula in a continuous state setting is novel.

*Remark* 2.7 (Interpretation as effective learning rate)**.** As for the expected return in Proposition 2.4, we note that the expected stopping time appears as a scaling factor in the state-space-based policy gradient formula (16). While one could be tempted to argue that this acts only as a constant multiplicative factor that can be absorbed into the learning rate in gradient based optimization (as, e.g., argued in Sutton & Barto (2018)), we note that $\mathbb{E}_\pi[N+1]$ depends on the current policy and can therefore substantially vary during the course of optimization. Conversely, omitting the factor leads

---

[5]In fact, assuming $N$ to be deterministic yields the same trajectory-based formula as in (15) and it is remarkable that the dependency of $N$ on $\pi$ in the random stopping time case does not add any extra term.

to a different *effective learning rate* in stochastic gradient ascent. To be precise, let $\{\gamma^{(k)}\}_k$ be a sequence of learning rates, then we may update the parameters via

$$\theta^{(k+1)} = \theta^{(k)} + \gamma^{(k)} \nabla_\theta J(\pi_{\theta^{(k)}}). \tag{19}$$

If we instead update

$$\theta^{(k+1)} = \theta^{(k)} + \gamma^{(k)} \nabla_\theta \widetilde{J}(\pi_{\theta^{(k)}}) \tag{20a}$$

$$= \theta^{(k)} + \widetilde{\gamma}^{(k)} \nabla_\theta J(\pi_{\theta^{(k)}}), \tag{20b}$$

where

$$\nabla_\theta \widetilde{J} := \nabla_\theta J / \mathbb{E}_{\pi_{\theta^{(k)}}}[N+1] \tag{21}$$

is the (wrong) gradient with omitted scaling factor, then $\widetilde{\gamma}^{(k)} = \gamma^{(k)} / \mathbb{E}_{\pi_{\theta^{(k)}}}[N+1]$ is the effective learning rate relating to gradient ascent relying on the correct gradient. We note that this learning rate may substantially vary with changing expected runtimes in the course of the optimization and refer to Section 3 for experiments investigating its effect on the optimization performance. Importantly, we note that previous policy gradient theorems have either stated the version without expected stopping time, i.e. have considered (21) for finite time scenarios, or only dealt with the infinite time horizon case (Sutton et al., 1999). Both cases lead to incorrect formulas when being applied to random time horizon problems.

For the trajectory-based policy gradient estimator (15) we can additionally derive the following alternative versions, which might be advantageous from a computational perspective and are proved in Appendix C.

**Corollary 2.8** (Alternative trajectory-based versions of the policy gradient)**.** *For the gradient of the expected return* (5) *it holds*

$$\nabla_\theta J(\pi) = \mathbb{E}_\pi \left[ \sum_{n=0}^N \nabla_\theta \log \pi_\theta(S_n, A_n) \sum_{m=0}^N r(S_m, A_m) \right] \tag{22}$$

$$= \mathbb{E}_\pi \left[ \sum_{n=0}^N \nabla_\theta \log \pi_\theta(S_n, A_n) \sum_{m=n}^N r(S_m, A_m) \right] \tag{23}$$

$$= \mathbb{E}_\pi \left[ \sum_{n=0}^N \nabla_\theta \log \pi_\theta(S_n, A_n) \left( Q^\pi(S_n, A_n) - b(S_n) \right) \right], \tag{24}$$

*where $b : \mathcal{S} \to \mathbb{R}$ is an arbitrary function (sometimes called baseline).*

For deterministic policies, the policy gradient takes a slightly different form, which we state in the following. We note that for infinite time horizons it has already been derived in Silver et al. (2014), however, we have not seen a formula for random time horizons before.

**Proposition 2.9** (Policy gradient for deterministic policies)**.** *For the gradient of the expected return* (5) *it holds*[6]

$$\nabla_\theta J(\mu_\theta) = \mathbb{E}_\mu \left[ \sum_{n=0}^{N} \nabla_\theta \mu_\theta(S_n)^\top \nabla_a Q^{\mu_\theta}(S_n, a)\Big|_{a=\mu_\theta(S_n)} \right] \tag{25}$$

$$= \mathbb{E}_\mu[N+1]\, \mathbb{E}_{s\sim\rho^\mu} \left[ \nabla_\theta \mu_\theta(s)^\top \nabla_a Q^{\mu_\theta}(s, a)\Big|_{a=\mu_\theta(s)} \right], \tag{26}$$

*where $\rho^\mu$ is the state-space density defined in* (8) *(with deterministic instead of stochastic policy).*

For the deterministic policy gradient we can further derive a version in the model-based setting, where we assume the knowledge of the transition density $p$ defined in (3).

**Corollary 2.10** (Model-based policy gradient for deterministic policies)**.** *For the gradient of the expected return* (5) *it holds*

$$\nabla_\theta J(\mu_\theta) = \mathbb{E}_\mu \left[ \sum_{n=0}^{N} \nabla_\theta \mu_\theta(S_n)^\top \Big( \nabla_a r(S_n, a) \right.$$
$$\left. + V^\mu(S_{n+1})\nabla_a \log p(S_{n+1}, S_n, a) \Big)\Big|_{a=\mu_\theta(S_n)} \right] \tag{27}$$

$$= \mathbb{E}_\mu[N+1]\, \mathbb{E}_{\substack{s\sim\rho^\mu,\\ s'\sim p^\mu(\cdot,s)}} \left[ \nabla_\theta \mu_\theta(s)^\top \Big( \nabla_a r(s, a) \right.$$
$$\left. + V^\mu(s')\nabla_a \log p(s', s, a) \Big)\Big|_{a=\mu_\theta(s)} \right]. \tag{28}$$

We note that the alternative versions stated in (22) and (23) for stochastic policies hold for deterministic policies only in the model-based and not in the model-free case, see Corollary C.5 in the appendix.

*Remark* 2.11 (Relation to stochastic optimal control). The trajectory-based formula (27) is equivalent to the control cost gradient in stochastic optimal control problems, e.g. stated in Ribera Borrell et al. (2024), when using certain Gaussian transition densities, see also Remark C.4 in Appendix C. We also note that in Quer & Ribera Borrell (2024) a version of (27) is heuristically derived after assuming $N$ to be deterministic. To the best of our knowledge, the presented model-based policy gradient formula with random time horizons has not been proved in the discrete time setting yet.

# 3. Numerical experiments

In this section we compare the previously discussed gradient estimators on numerical examples. In particular, we

compare trajectory and state-space based formulas and investigate the effect of neglecting the factor $\mathbb{E}_\pi[N+1]$ in the policy gradient (PG) computations. To be precise, for all our experiments we compute the gradient either with the trajectory-based expression (15) (*trajectory PG*), by the state-space expression (16) (*state-space PG*), or by the modified state-space expression (21), which omits the scaling factor and is therefore, strictly speaking, an incorrect gradient (*state-space PG (biased)*). We refer to Remark 2.7 for interpreting this incorrect formula as an effective learning rate that changes over the course of optimization. Crucially, note that this effective learning rate is highly problem-specific and can not be controlled easily – this is an obvious downside of the gradient (21). In order to assure fair comparisons, we first search for the optimal learning rate for each gradient approach in all of our experiments and use standard stochastic gradient ascent for optimization (in order to not have interacting effects with sophisticated optimization methods). We refer to Algorithms 1-4 in Appendix D for further computational details. The code can be found at `https://github.com/riberaborrell/rl-random-times`.

## 3.1. Modified continuous mountain car problem

The mountain car problem is a classical benchmark in reinforcement learning, where the goal is to reach the top of a mountain by leveraging gravitational energy additional to the car's acceleration (Singh & Sutton, 1996). We consider the state space $\mathcal{S} = \mathcal{D}_{\text{pos}} \times \mathcal{D}_{\text{vel}} \subset \mathbb{R}^2$, where $\mathcal{D}_{\text{pos}} = [-1.2, 0.6]$, $\mathcal{D}_{\text{vel}} = [-0.07, 0.07]$ and the action space $\mathcal{A} = [-1, 1] \subset \mathbb{R}$. The target set is defined as $\mathcal{T} = [0.45, \infty) \times \mathbb{R}$ and the deterministic dynamics[7] is described via the function $h : \mathcal{S} \times \mathcal{A} \to \mathcal{S}$, given by

$$\begin{aligned} v' &= v + 0.0015\,a - 0.0025\cos(3x), \\ x' &= x + v', \end{aligned} \tag{29}$$

where the state variable $s = (x, v)^\top$ has a position and a velocity part. Compared to the typical problem, we consider trajectories that only stop when reaching the target set and make the problem slightly harder as we aim for reaching this goal quickly by integrating the runtime into the reward function,

$$r(s, a) := \begin{cases} -1 - 0.1a^2 & \text{if } s \notin \mathcal{T}, \\ 0 & \text{if } s \in \mathcal{T}. \end{cases} \tag{30}$$

We consider a Gaussian stochastic policy whose mean and covariance are given by a feed-forward neural network with 3 layers and 32 hidden units per layer. We refer to Appendix E.2 for further details. In Figure 2 we compare the

---

[6]We denote with $\nabla_\theta \mu_\theta$ the Jacobian matrix of $\mu_\theta$.

[7]We can think of the transition density defined in (3) as a Dirac delta distribution $p(s', s, a) = \delta(s' - h(s, a))$.

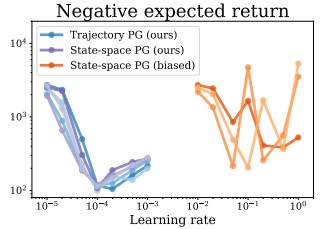 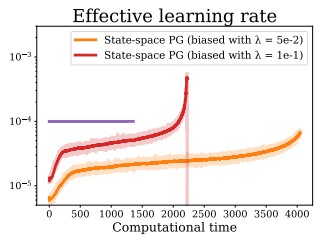 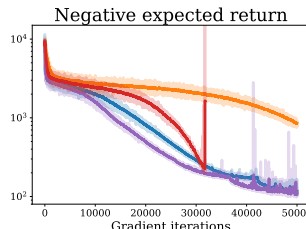 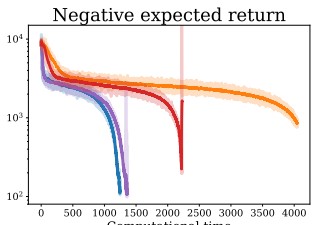

*Figure 2.* We display the performance of the three different policy gradients (PG) on the mountain car problem described in Section 3.1. In the left plot, the negative expected return is investigated for different learning rates, the different transparency values indicate different runs. For the performance plots, we always choose the best respective learning rate. The second plot shows the effective learning rates, see Remark 2.7 for an explanation (we always display a moving average above the raw data). We see that for the biased methods, the effective learning rate increases significantly over the course of the optimization. The two plots on the right-hand side display the negative expected return – once per gradient step and once per computational time (in minutes). We see that the PG formulas incorporating the expected hitting time perform significantly better. For the biased formula, the algorithm stops at some point due to increased trajectory lengths and related memory issues.

three different policy gradient formulas and see that the ones incorporating the expected stopping time perform significantly better. This can be explained with the effective learning rate of the biased method, which increases significantly during the optimization due to the change of the expected hitting times (from $N \approx 10^4$ to $N \approx 10^2$). We highlight that this behavior is problem-specific and cannot be known a priori.

### 3.2. Two-joint robot arm (reacher)

The reacher environment contains a two-joint robot arm whose goal is to move its fingertip close to the target. It operates on continuous state and action spaces $\mathcal{S} = [-1,1]^4 \times \mathbb{R}^6 \subset \mathbb{R}^{10}$, $\mathcal{A} = [-1,1]^2 \subset \mathbb{R}^2$. The dynamics is deterministic and the initial state is sampled, see Towers et al. (2024) for details. Since the original reacher environment is posed for infinite time horizons, we define the target set such that trajectories terminate if the positions $s_9, s_{10}$ of the fingertip are near the target and the angular velocities $s_7, s_8$ of the two arms are low enough,

$$\mathcal{T} = \{s \in \mathcal{S} : \|(s_7, s_8)\| \leq 2, \|(s_9, s_{10})\| \leq 0.05\}. \quad (31)$$

We consider the (slightly modified) reward function

$$r(s,a) := \begin{cases} -1 - \omega_{\text{control}}\|a\|^2 & \text{if} \quad s \notin \mathcal{T}, \\ 0 & \text{if} \quad s \in \mathcal{T}, \end{cases} \quad (32)$$

where $\omega_{\text{control}} = 0.1$, again including a penalty for long runtimes as to motivate the robot to operate quickly, however, with small actions. We again consider a Gaussian stochastic policy, see Appendix E.3 for details. In Figure 3 we can see that our PG formulas lead to faster convergence compared to the classical (biased) formula.

### 3.3. Importance sampling of hitting times in molecular dynamics

Finally, we consider a problem that is relevant for the estimation of rare events in the context of molecular dynamics (Hartmann et al., 2017). The goal is to identify a deterministic policy that allows for the effective simulation of reaching a target set $\mathcal{T} \subset \mathcal{S}$ by shortening trajectories and reducing the variance of corresponding Monte Carlo estimators (Hartmann & Schütte, 2012). To this end, we consider $d$-dimensional state and action spaces $\mathcal{S} = \mathcal{A} = \mathbb{R}^d$, a Gaussian transition density defined by

$$p(\cdot, s, a) = \mathcal{N}\left(s + (a - \nabla U(s))\,\Delta t, \sigma^2 \Delta t\,\text{Id}\right), \quad (33)$$

where $U : \mathbb{R}^d \to \mathbb{R}$ is a so-called potential function that is given by the specific problem and $\Delta t > 0$ is a step size. The reward function is given by

$$r(s,a) = \begin{cases} -\Delta t - \frac{1}{2}\|a\|^2 \Delta t & \text{if} \quad s \notin \mathcal{T}, \\ 0 & \text{if} \quad s \in \mathcal{T}. \end{cases} \quad (34)$$

Intuitively, (34) can be interpreted as aiming to minimize the runtime $N$, while keeping the actions small. This problem has already been studied with reinforcement learning methods in Quer & Ribera Borrell (2024). In our experiments we consider $U(s) = \sum_{i=1}^{d} \alpha_i (s_i^2 - 1)^2$, where $\alpha_i$ quantifies the amount of metastability in the $i$-th dimension, as well as $\sigma = \sqrt{2}$ and $\Delta t = 10^{-2}$. For the gradient computations we rely on the model-based formulas stated in Corollary 2.10.

We consider $d = 20$ and choose $\alpha_1 = 5$, $\alpha_2 = 2$, $\alpha_i = 0.5$, for $i = \{3, \ldots, 20\}$ as well as $\mathcal{T} = \widetilde{\mathcal{T}} \times \mathbb{R}^{d-2}$, where

$$\widetilde{\mathcal{T}} = \{(s_1, s_2) \in \mathcal{S} : s_1, s_2 > 0, U(s) \leq 0.25\}, \quad (35)$$

see also Figure 1 for an illustration. As described above, we first search for a suitable learning rate for each approach. In the first panel in Figure 4 we plot the performance dependence on the learning rate. As expected, the trajectory

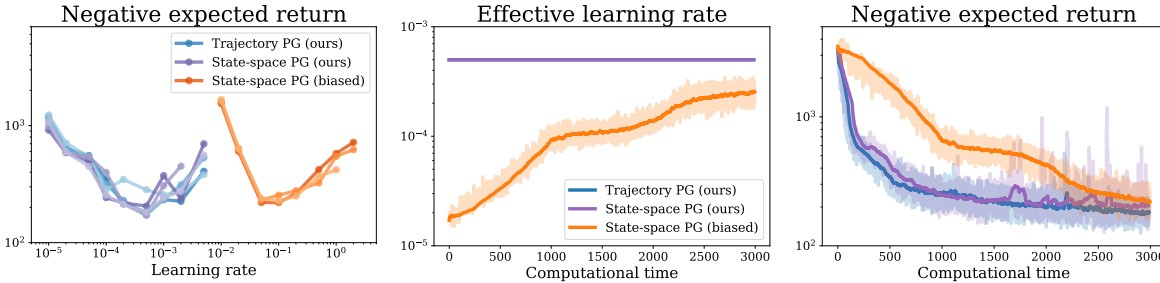

*Figure 3.* We display the performance of the three different policy gradients (PG) on the reacher problem described in Section 3.2. As before, the left plot shows the performance w.r.t. to different learning rates, from which we choose the best learning rate for each method. In the second plot we can see that the effective learning rate for the biased state-space PG, which ignores the expected hitting time, is rather small at the beginning and increases over the course of the optimization. This then results in slower convergence to the optimal performance, as can be seen in the plot on the right-hand side.

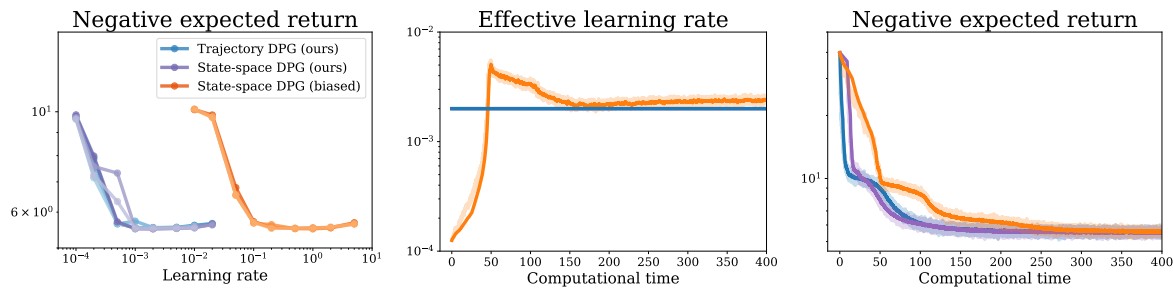

*Figure 4.* We consider importance sampling of hitting times in molecular dynamics as described in Section 3.3 and compare the three different deterministic policy gradients (DPG). We display the performance depending on the learning rate as well as the effective learning rate and the negative expected return over the course of the optimization. We see that biased state-spaced PG converges more slowly compared to our DPG methods.

and state-space based gradient computations yield similar results – they only differ due to statistical effects. The scaled state-space gradient, on the other hand, yields very different results. For each approach, we choose the optimal learning rate. As noted in Remark 2.7, the scaled state-space approach has an effective learning rate that depends on the current expected stopping time. In the second panel we plot this quantity over the course of the optimization. Since the stopping times get shorter during the optimization, the effective learning rate increases. Consequently, this leads to slower optimization, since especially in the beginning the effective learning rate is comparatively small, see the third panel. In fact, the formulas that incorporate the stopping time (and thus provide the correct gradient) converge roughly twice as fast.

## 4. Conclusion and outlook

In this work, we have suggested a theoretical framework for reinforcement learning problems with random time horizons. In our experiments, we have seen that the related (mostly novel) policy gradient formulas can lead to improved and accelerated performance. Crucially, we can explain those

performance improvements with the formulas we derived. We anticipate that our framework will allow to design advanced optimization algorithms that cleverly integrate the random time dependency, potentially combined with off-policy algorithms, leading to further improvements on state-of-the-art problems in the future. In particular, we envision integrations of our corrections for random time horizons into methods such as *trust region policy optimization* (Schulman et al., 2015) and *proximal policy optimization* (Schulman et al., 2017). Furthermore, we think that the connection of reinforcement learning to (stochastic) optimal control problems – studying e.g. continuous-time perspectives (Wang et al., 2020), path space measures (Nüsken & Richter, 2021) or diffusion models (Berner et al., 2024) – will be fruitful both for further theoretical insights and novel numerical algorithms.

## Acknowledgments

The first author would like to thank A. Sikorski for his helpful insights on understanding the infinite horizon setting as a random time problem and previous fruitful discussions. This research has been partially funded by Deutsche Forschungsgemeinschaft (DFG) through grant CRC 1114 (Project No. 235221301) and under Germany's Excellence Strategy MATH+: Berlin Mathematics Research Center (EXC 2046/1, Project No. 390685689).

## Impact statement

The goal of this work is to advance the theoretical understanding of the field of reinforcement learning, leading to improvements in applications as well. While there are potential societal consequences of our work in principle, we do not see any concrete issues and thus believe that we do not specifically need to highlight any.

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

## A. Notation and setting

As stated in Section 2, we denote with $\mathcal{M} = (\mathcal{S}, \mathcal{A}, r_n, \rho_0, p)$ a time-discrete *Markov decision process*, where $\mathcal{S} \subset \mathbb{R}^{d_s}$ is the state space, $\mathcal{A} \subset \mathbb{R}^{d_a}$ is the action space, $r_n$ is the reward function, $\rho_0$ is the initial probability density of the states (defined in (2)) and $p$ is the time-homogeneous (state-action) transition density (defined in (3)). Further, we consider the stochastic policy $\pi$ or the deterministic policy $\mu$. The dynamics of the Markov decision process is defined via $S_0 \sim \rho_0$ as well as $A_n \sim \pi(S_n, \cdot)$ and $S_{n+1} \sim p(\cdot, S_n, A_n)$ for each $n \in \{0, \dots, N-1\}$, and we call the resulting set $\{S_0, A_0, S_1, A_1, \dots, S_N\}$ a state-action trajectory and $\{S_0, \dots, S_N\}$ a (state) trajectory.

For the expectation operator we write $\mathbb{E}_\pi$ if the random variables in the expectation depend on the policy $\pi$, however note that the randomness not only depends on $\pi$, but also on the initial density $\rho_0$ and the transition density $p$, i.e. for a function $\varphi : (\mathcal{S} \times A)^N \times \mathcal{S} \to \mathcal{S}$ and random variables $S_0, A_0, \dots, S_N$, and for fixed $N$, we write

$$\mathbb{E}_\pi [\varphi(S_0, A_0, \dots, S_N)] = \mathbb{E}_{\rho_0, p, \pi} [\varphi(S_0, A_0, \dots, S_N)] \tag{36a}$$

$$= \int_{(\mathcal{S} \times \mathcal{A})^N \times \mathcal{S}} \rho_0(s_0) \left( \prod_{n=0}^{N-1} \pi(s_n, a_n) p(s_{n+1}, s_n, a_n) \right) \mathrm{d}s_0 \mathrm{d}a_0 \dots \mathrm{d}s_{N-1} \mathrm{d}a_{N-1} \mathrm{d}s_N. \tag{36b}$$

In our work, the stopping $N$ is typically considered to be random, so the expectation is also over $N$, i.e.

$$\mathbb{E}_\pi [\varphi(S_0, A_0, \dots, S_N)] = \mathbb{E}_{\rho_0, p, \pi, N} [\varphi(S_0, A_0, \dots, S_N)] \tag{37a}$$

$$= \sum_{m=0}^{\infty} \mathbb{P}(N = m) \int_{(\mathcal{S} \times \mathcal{A})^m \times \mathcal{S}} \rho_0(s_0) \left( \prod_{n=0}^{m-1} \pi(s_n, a_n) p(s_{n+1}, s_n, a_n) \right) \mathrm{d}s_0 \mathrm{d}a_0 \dots \mathrm{d}s_{m-1} \mathrm{d}a_{m-1} \mathrm{d}s_m, \tag{37b}$$

where $\mathbb{P}(N = m)$ is the probability of $N$ having the value $m$ and we use the convention $\prod_{n=0}^{-1}(\cdot)_n = 1$. This probability is typically dependent on the dynamics, and therefore the policy, and is thus unknown. For deterministic policies we write $\mathbb{E}_\mu$ (even though $\mu : \mathcal{S} \to \mathcal{A}$ is a deterministic function). Also, note that for state-space expectations stated e.g. in (11) we write $\mathbb{E}_{\substack{s \sim \rho^\pi \\ a \sim \pi(s, \cdot)}} [\cdot]$ to denote $\mathbb{E}_{s \sim \rho^\pi} [\mathbb{E}_{a \sim \pi(s, \cdot)}[\cdot]]$.

Further, we assume that $\pi$ or $\mu$ are parametrized by the parameter vector $\theta \in \mathbb{R}^p$, and we may interchangeably write $\pi = \pi_\theta$ or $\mu = \mu_\theta$ for ease of notation. We write $\nabla_\theta$ to denote the gradient w.r.t. the parameter vector $\theta$ and for any $s \in \mathcal{S}$ we write $\nabla_\theta \mu_\theta(s)$ to denote the Jacobian matrix of $\mu_\theta(s)$ w.r.t. $\theta$.

## B. Additional background on reinforcement learning

Let us consider the (state-action) *reward function*[8] $r_n : \mathcal{S} \times \mathcal{A} \to \mathbb{R}$ and recall that it provides the signal that is received after being in state $S_n \in \mathcal{S}$ and having taken action $A_n \in \mathcal{A}$ at time $n \in \mathbb{N}$. In our work we consider a time-independent reward function and we denote it by $r$. We define the *return (from step $n$ onward)* by

$$G_n = \sum_{m=n}^{N} r(S_m, A_m) \tag{38}$$

and denote $G_0$ by the *(initial) return*. We define the *expected (initial) return* (sometimes also called objective function) as

$$J(\pi) = \mathbb{E}_\pi \left[ \sum_{n=0}^{N} r(S_n, A_n) \right]. \tag{39}$$

We note that starting at $n = 0$ holds without loss of generality and one could also start at any other time, since the problem is time-autonomous. Further, we can define the *value function* (or *return-to-go*) as the expected return conditioned on starting at state $s \in \mathcal{S}$,

$$V^\pi(s) := \mathbb{E}_\pi \left[ \sum_{n=0}^{N} r(S_n, A_n) \middle| S_0 = s \right]. \tag{40}$$

---

[8]In principle, the reward function can also depend on the following next state.

We can further condition on applying the action $a \in \mathcal{A}$, and define the so-called *Q-value function* as

$$Q^\pi(s, a) := \mathbb{E}_\pi \left[ \sum_{n=0}^{N} r(S_n, A_n) \middle| S_0 = s, A_0 = a \right]. \tag{41}$$

We note that it holds $J(\pi) = \mathbb{E}_{s \sim \rho_0}[V^\pi(s)]$ and $V^\pi(s) = \mathbb{E}_{a \sim \pi(s, \cdot)}[Q^\pi(s, a)]$. For all expressions we have the Bellman equation[9], e.g.

$$J(\pi) = \mathbb{E}_\pi \left[ r(S_0, A_0) + \sum_{n=1}^{N} r(S_n, A_n) \right] \tag{42a}$$

$$= \mathbb{E}_\pi \left[ r(S_0, A_0) + \mathbb{E}_\pi \left[ \sum_{n=1}^{N} r(S_n, A_n) \middle| S_1 \right] \right] \tag{42b}$$

$$= \mathbb{E}_\pi \left[ r(S_0, A_0) + V^\pi(S_1) \right]. \tag{42c}$$

Analogously, it holds for all $s \in \mathcal{T}^c$

$$V^\pi(s) = \mathbb{E}_\pi \left[ r(S_0, A_0) + V^\pi(S_1) | S_0 = s \right], \tag{43a}$$

$$= \int_\mathcal{A} \pi(s, a) \left( r(s, a) + \int_\mathcal{S} p(s', s, a) V^\pi(s') \mathrm{d}s' \right) \mathrm{d}a, \tag{43b}$$

and

$$Q^\pi(s, a) = r(s, a) + \mathbb{E}_\pi \left[ Q^\pi(S_1, A_1) | S_0 = s, A_0 = a \right] \tag{44a}$$

$$= r(s, a) + \int_\mathcal{S} p(s', s, a) \int_\mathcal{A} \pi(s', a') Q^\pi(s', a') \mathrm{d}a' \mathrm{d}s'. \tag{44b}$$

For deterministic policies the above equations reduce to

$$V^\mu(s) = r(s, \mu(s)) + \mathbb{E}_\mu \left[ V^\mu(S_1) | S_0 = s \right] \tag{45a}$$

$$= r(s, \mu(s)) + \int_\mathcal{S} p(s', s, \mu(s)) V^\mu(s') \mathrm{d}s', \tag{45b}$$

and

$$Q^\mu(s, a) = r(s, a) + \mathbb{E}_\mu \left[ Q^\mu(S_1, \mu(S_1)) | S_0 = s, A_0 = a \right] \tag{46a}$$

$$= r(s, a) + \int_\mathcal{S} p(s', s, a) Q^\mu(s', \mu(s')) \mathrm{d}s'. \tag{46b}$$

$$= r(s, a) + \int_\mathcal{S} p(s', s, a) V^\mu(s') \mathrm{d}s'. \tag{46c}$$

Further, note that in control theory the *value function* denotes the *optimal cost-to-go*, i.e. the optimal value of the cost functional w.r.t. all possible controls. In contrast, in reinforcement learning the term *value function* is used for any arbitrary policy. If the policy is optimal it is called *optimal value function*.

In practice, one often chooses $\pi = \mathcal{N}(\mu, \sigma^2 \operatorname{Id})$, for which $\mu$ and $\sigma$ are learnable functions. We note that with $\sigma$ approaching the zero function $x \mapsto 0$, i.e. with reducing the stochasticity of the stochastic policy, $\pi$ approaches a Dirac distribution and therefore a deterministic policy, see also Remark 2.1 in the main text and Theorem 2 in Silver et al. (2014).

## C. Proofs and additional statements

*Proof of Lemma 2.2.* First note the following re-ordering of sums, containing the sequences $\{a_i\}_i$ and $\{b_j\}_j$. It holds that

$$\sum_{i=0}^{\infty} a_i \sum_{j=0}^{i} b_j = \lim_{l \to \infty} \sum_{i=0}^{l} a_i \sum_{j=0}^{i} b_j = \lim_{l \to \infty} \sum_{j=0}^{l} b_j \sum_{i=j}^{l} a_i = \sum_{j=0}^{\infty} b_j \sum_{i=j}^{\infty} a_i. \tag{47}$$

---

[9]Note that for the infinite time horizon case the value function is scaled by the discount factor $\gamma$.

Assuming that $N_\gamma \sim \mathrm{Geom}(1 - \gamma)$ with $\gamma \in (0, 1)$, we have that $\mathbb{P}(N_\gamma = m) = \gamma^m(1 - \gamma)$ and can compute

$$\mathbb{E}_\pi \left[ \sum_{n=0}^{N_\gamma} r(S_n, A_n) \right] = \mathbb{E}_{N_\gamma} \left[ \mathbb{E}_\pi \left[ \sum_{n=0}^{N_\gamma} r(S_n, A_n) \right] \right] \tag{48a}$$

$$= \sum_{m=0}^{\infty} \mathbb{P}(N_\gamma = m) \mathbb{E}_\pi \left[ \sum_{n=0}^{m} r(S_n, A_n) \right] \tag{48b}$$

$$= \mathbb{E}_\pi \left[ \sum_{n=0}^{\infty} r(S_n, A_n)(1 - \gamma) \sum_{m=n}^{\infty} \gamma^m \right] \tag{48c}$$

$$= \mathbb{E}_\pi \left[ \sum_{n=0}^{\infty} \gamma^n r(S_n, A_n) \right], \tag{48d}$$

where we used the tower property in the first line, the fact that $N_\gamma$ does not depend on the policy in (48b) and identity (47) in (48c). We can further check that

$$\mathbb{P}(N_\gamma = \infty) = \lim_{m \to \infty} \mathbb{P}(N_\gamma = m) = \lim_{m \to \infty} \gamma^m(1 - \gamma) = 0. \tag{49}$$

$\square$

*Proof of Lemma 2.3.* Recalling the definitions (7) and (8),

$$\eta^\pi := \sum_{n=0}^{\infty} \rho_n^\pi, \qquad \int_\Lambda \rho_n^\pi(s) \mathrm{d}s = \mathbb{P}_\pi(S_n \in \Lambda), \tag{50}$$

we compute

$$\int_\Lambda \eta^\pi(s) \mathrm{d}s = \sum_{n=0}^{\infty} \int_\Lambda \rho_n^\pi(s) \mathrm{d}s = \sum_{n=0}^{\infty} \mathbb{P}_\pi(S_n \in \Lambda) = \sum_{n=0}^{\infty} \mathbb{E}_\pi \left[ \mathbb{1}_\Lambda(S_n) \right] = \mathbb{E}_\pi \left[ \sum_{n=0}^{\infty} \mathbb{1}_\Lambda(S_n) \right] = \mathbb{E}_\pi \left[ \sum_{n=0}^{N} \mathbb{1}_\Lambda(S_n) \right]. \tag{51}$$

Further, we note that choosing $\Lambda = \mathcal{S}$ yields

$$Z^\pi = \int_{\mathcal{S}} \eta^\pi(s) \mathrm{d}s = \mathbb{E}_\pi[N] + 1. \tag{52}$$

$\square$

*Proof of Proposition 2.4.* Let us first define the $n$-step transition function $p_n^\pi(s', s)$ via

$$\int_\Lambda p_n^\pi(s', s) \mathrm{d}s' = \mathbb{P}_\pi(S_n \in \Lambda | S_0 = s). \tag{53}$$

Note that we have

$$\int_{\mathcal{S}} p_n^\pi(s', s) \mathrm{d}s' = \mathbb{P}_\pi(S_n \in \mathcal{S} | S_0 = s) = \mathbb{P}_\pi(S_n \in \mathcal{S}, n \leq N | S_0 = s) = \mathbb{P}_\pi(n \leq N | S_0 = s), \tag{54}$$

so it is only a density conditioned on the fact that $n \leq N$. We can now compute

$$\mathbb{E}_\pi \left[ \sum_{n=0}^N r(S_n, A_n) \right] = \mathbb{E}_\pi \left[ \sum_{n=0}^\infty \mathbb{1}_{[n,\infty)}(N) r(S_n, A_n) \right] \tag{55a}$$

$$= \int_\mathcal{S} \int_\mathcal{S} \int_\mathcal{A} \sum_{n=0}^\infty r(s,a)\pi(s,a) p_n^\pi(s,\bar{s}) \rho_0(\bar{s}) \, \mathrm{d}\bar{s} \, \mathrm{d}s \, \mathrm{d}a \tag{55b}$$

$$= \int_\mathcal{S} \int_\mathcal{A} r(s,a)\pi(s,a) \sum_{n=0}^\infty \int_\mathcal{S} p_n^\pi(s,\bar{s}) \rho_0(\bar{s}) \, \mathrm{d}\bar{s} \, \mathrm{d}s \, \mathrm{d}a \tag{55c}$$

$$= \int_\mathcal{S} \int_\mathcal{A} r(s,a)\pi(s,a) \sum_{n=0}^\infty \rho_n^\pi(s) \, \mathrm{d}s \, \mathrm{d}a \tag{55d}$$

$$= \int_\mathcal{S} \int_\mathcal{A} r(s,a)\pi(s,a) \eta^\pi(s) \, \mathrm{d}s \, \mathrm{d}a \tag{55e}$$

$$= \mathbb{E}_\pi[N+1] \mathbb{E}_{\substack{s \sim \rho^\pi \\ a \sim \pi(s,\cdot)}} [r(s,a)], \tag{55f}$$

where in the last line we used

$$\eta^\pi = \rho^\pi \int_\mathcal{S} \eta^\pi(s)\mathrm{d}s = \rho^\pi \, \mathbb{E}_\pi[N+1] \tag{56}$$

via definition (8) and Lemma 2.3. Note that the sum in (55b) switches from a sum until the (potentially random) time $N$ to an infinite sum since the transition probability $p_n^\pi$ defined in (54) encodes not only the states, but also a potential stopping of corresponding trajectories and thus encodes the potential randomness of $N$. □

*Remark* C.1 (Alternative derivation of state-space expected return). The state-space based formula for the expected return stated in Proposition 2.4 can also be derived as follows. The trajectory-based formula

$$J(\pi) = \mathbb{E}_\pi \left[ \sum_{n=0}^N r_n(S_n, A_n) \right], \tag{57}$$

stated in (5), can be approximated by $K$ trajectories via

$$\frac{1}{K} \sum_{k=1}^K \sum_{n=0}^{N^{(k)}} r(S_n^{(k)}, A_n^{(k)}), \tag{58}$$

noting that each trajectory has a different length, namely $N^{(k)} + 1$ steps in the $k$-th trajectory. We note that we can consider the particles

$$B := \bigcup_{k=1}^K \bigcup_{n=0}^{N^{(k)}} \left\{ S_n^{(k)}, A_n^{(k)} \right\} \tag{59}$$

as an unbiased sample from the action-state-space density $\rho^\pi(s)\pi(s,a)$, defined in (8). Let us merge the two indices of the elements of $B$ and write

$$B = \bigcup_{m=1}^M \left\{ \widetilde{S}^{(m)}, \widetilde{A}^{(m)} \right\}, \tag{60}$$

noting that $M = \sum_{k=1}^K (N^{(k)} + 1)$. We can now write the sample estimator of the trajectory-based expected return (58) as

$$\frac{1}{K} \sum_{k=1}^K \sum_{n=0}^{N^{(k)}} r(S_n^{(k)}, A_n^{(k)}) = \frac{1}{K} \sum_{m=1}^M r(\widetilde{S}^{(m)}, \widetilde{A}^{(m)}) = \frac{\frac{M}{K}}{M} \sum_{m=1}^M r(\widetilde{S}^{(m)}, \widetilde{A}^{(m)}). \tag{61}$$

Now, noting that $M/K$ converges to $\mathbb{E}_\pi[N+1]$ for $K \to \infty$ by the law of large numbers, we conclude that (61) converges to the state-space expected return

$$J(\pi) = \mathbb{E}_\pi[N+1] \mathbb{E}_{\substack{s \sim \rho^\pi \\ a \sim \pi(s,\cdot)}} [r(s,a)], \tag{62}$$

as stated in (11).

**Lemma C.2** (Unrolling for stochastic policies). *For any $l \geq 1$ it holds that*

$$\nabla_\theta V^\pi(s) = \sum_{n=0}^{l-1} \int_{\mathcal{S}} p_n^\pi(s', s) \int_{\mathcal{A}} \nabla_\theta \pi_\theta(s', a) Q^\pi(s', a) \mathrm{d}a \, \mathrm{d}s' + \int_{\mathcal{S}} p_l^\pi(s', s) \nabla_\theta V^\pi(s') \mathrm{d}s'. \tag{63}$$

*Proof.* For the sake of notational convenience, let us define the shorthand notation

$$h_\theta(s, a) := \nabla_\theta \pi_\theta(s, a) Q^\pi(s, a). \tag{64}$$

We prove Lemma C.2 via induction over $l$. Let us start by the initial case $l = 1$. Recalling that $V^\pi(s) = \mathbb{E}_{a \sim \pi(s, \cdot)}[Q^\pi(s, a)]$, we can compute

$$\nabla_\theta V^\pi(s) = \int_{\mathcal{A}} \Big( \nabla_\theta \pi_\theta(s, a) Q^\pi(s, a) + \pi_\theta(s, a) \nabla_\theta Q^\pi(s, a) \Big) \mathrm{d}a \tag{65a}$$

$$= \int_{\mathcal{A}} h_\theta(s, a) \mathrm{d}a + \int_{\mathcal{A}} \pi_\theta(s, a) \nabla_\theta \Big( r(s, a) + \int_{\mathcal{S}} p(s', s, a) V^\pi(s') \mathrm{d}s' \Big) \mathrm{d}a \tag{65b}$$

$$= \int_{\mathcal{A}} h_\theta(s, a) \mathrm{d}a + \int_{\mathcal{A}} \pi_\theta(s, a) \int_{\mathcal{S}} p(s', s, a) \nabla_\theta V^\pi(s') \mathrm{d}s' \mathrm{d}a \tag{65c}$$

$$= \int_{\mathcal{A}} h_\theta(s, a) \mathrm{d}a + \int_{\mathcal{S}} p_1^\pi(s', s) \nabla_\theta V^\pi(s') \mathrm{d}s', \tag{65d}$$

where we use the Bellman equation stated in (44) and where $p_1^\pi$ is defined in (53). The expression (65) coincides with (63) for $l = 1$. Let us now assume that (63) holds for $l - 1$. We can compute

$$\nabla_\theta V^\pi(s) = \sum_{n=0}^{l-2} \int_{\mathcal{S}} p_n^\pi(s', s) \int_{\mathcal{A}} h_\theta(s', a) \mathrm{d}a \, \mathrm{d}s' + \int_{\mathcal{S}} p_{l-1}^\pi(s', s) \nabla_\theta V^\pi(s') \mathrm{d}s' \tag{66a}$$

$$= \sum_{n=0}^{l-2} \int_{\mathcal{S}} p_n^\pi(s', s) \int_{\mathcal{A}} h_\theta(s', a) \mathrm{d}a \, \mathrm{d}s' + \int_{\mathcal{S}} p_{l-1}^\pi(s', s) \left( \int_{\mathcal{A}} h_\theta(s', a) \mathrm{d}a + \int_{\mathcal{S}} p_1^\pi(s'', s') \nabla_\theta V^\pi(s'') \mathrm{d}s'' \right) \mathrm{d}s' \tag{66b}$$

$$= \sum_{n=0}^{l-1} \int_{\mathcal{S}} p_n^\pi(s', s) \int_{\mathcal{A}} h_\theta(s', a) \mathrm{d}a \, \mathrm{d}s' + \int_{\mathcal{S}} \int_{\mathcal{S}} p_{l-1}^\pi(s', s) p_1^\pi(s'', s') \nabla_\theta V^\pi(s'') \mathrm{d}s' \mathrm{d}s'' \tag{66c}$$

$$= \sum_{n=0}^{l-1} \int_{\mathcal{S}} p_n^\pi(s', s) \int_{\mathcal{A}} h_\theta(s', a) \mathrm{d}a \, \mathrm{d}s' + \int_{\mathcal{S}} p_l^\pi(s'', s) \nabla_\theta V^\pi(s'') \mathrm{d}s'', \tag{66d}$$

where we have used (65) in the second line. Since (66d) conincides with (63) we have proved the statement by induction. $\square$

*Proof of Proposition 2.6.* We assume that the gradient of the value function with respect to the policy parameters is bounded, i.e., for any $\theta \in \mathbb{R}^p$ there exists an $L_\theta > 0$ such that

$$\sup_{s \in \mathcal{S}} ||\nabla_\theta V^\pi(s)|| \leq L_\theta. \tag{67}$$

Then it holds

$$\lim_{l \to \infty} \int_{\mathcal{S}} p_l^\pi(s', s) ||\nabla_\theta V^\pi(s')|| \mathrm{d}s' \leq \lim_{l \to \infty} L_\theta \, \mathbb{P}_\pi(l \leq N \mid S_0 = s) = 0, \tag{68}$$

by assumption of the (potentially random, but almost surely finite) time $N$. In the limit $l \to \infty$, the expression from Lemma C.2 therefore turns into

$$\nabla_\theta V^\pi(s) = \sum_{n=0}^{\infty} \int_{\mathcal{S}} p_n^\pi(s', s) \int_{\mathcal{A}} \nabla_\theta \pi_\theta(s', a) Q^\pi(s', a) \mathrm{d}a \, \mathrm{d}s'. \tag{69}$$

Further, we can compute

$$\nabla_\theta J(\pi_\theta) = \int_\mathcal{S} \rho_0(s_0) \nabla_\theta V^\pi(s) \mathrm{d}s_0 = \int_\mathcal{S} \rho_0(s_0) \sum_{n=0}^\infty \int_\mathcal{S} p_n^\pi(s, s_0) \int_\mathcal{A} \nabla_\theta \pi_\theta(s, a) Q^\pi(s, a) \mathrm{d}a \, \mathrm{d}s \, \mathrm{d}s_0 \tag{70a}$$

$$= \int_\mathcal{S} \eta^\pi(s) \int_\mathcal{A} \nabla_\theta \pi_\theta(s, a) Q^\pi(s, a) \mathrm{d}s \, \mathrm{d}a \tag{70b}$$

$$= \mathbb{E}[N+1] \int_\mathcal{S} \rho^\pi(s) \int_\mathcal{A} \pi_\theta(s, a) \nabla_\theta \log \pi_\theta(s, a) Q^\pi(s, a) \mathrm{d}s \, \mathrm{d}a \tag{70c}$$

$$= \mathbb{E}[N+1] \mathbb{E}_{\substack{s \sim \rho^\pi \\ a \sim \pi_\theta(s, \cdot)}} [\nabla_\theta \log \pi_\theta(s, a) Q^\pi(s, a)], \tag{70d}$$

where we have again used

$$\eta^\pi = \rho^\pi \int_\mathcal{S} \eta^\pi(s) \mathrm{d}s = \rho^\pi \, \mathbb{E}_\pi[N+1] \tag{71}$$

via definition (8) and Lemma 2.3. Finally, by choosing[10] $r(s, a) = \nabla_\theta \log \pi_\theta(s, a) Q^\pi(s, a)$ in Proposition 2.4 and using identity (70), we readily get

$$\nabla_\theta J(\pi_\theta) = \mathbb{E}_\pi \left[ \sum_{n=0}^N \nabla_\theta \log \pi_\theta(S_n, A_n) Q^\pi(S_n, A_n) \right]. \tag{72}$$

$\square$

*Proof of Corollary 2.8.* We first note that for an arbitrary function $\varphi : (\mathcal{S} \times \mathcal{A})^{n+1} \to \mathbb{R}$ it holds

$$\mathbb{E}_\pi [\varphi(S_0, A_0, \ldots, A_n, S_n) Q^\pi(S_n, A_n)] = \mathbb{E}_\pi \left[ \varphi(S_0, A_0, \ldots, A_n, S_n) \mathbb{E}_\pi \left[ \sum_{m=n}^N r(S_m, A_m) \middle| S_n, A_n \right] \right] \tag{73a}$$

$$= \mathbb{E}_\pi \left[ \mathbb{E}_\pi \left[ \varphi(S_0, A_0, \ldots, A_n, S_n) \sum_{m=n}^N r(S_m, A_m) \right] \right] \tag{73b}$$

$$= \mathbb{E}_\pi \left[ \varphi(S_0, A_0, \ldots, A_n, S_n) \sum_{m=n}^N r(S_m, A_m) \right], \tag{73c}$$

where we used the tower property, the fact that $\varphi(S_0, A_0, \ldots, S_n, A_n)$ is $\mathcal{F}_n$-measurable and the Markov property. Equality (23) now follows from choosing $\varphi(S_0, A_0 \ldots, S_n, A_n) = \nabla_\theta \log \pi(S_n, A_n)$ and summing over $n$. For this choice we can even show that

$$\mathbb{E}_\pi [\nabla_\theta \log \pi(S_n, A_n) Q^\pi(S_n, A_n)] = \mathbb{E}_\pi \left[ \nabla_\theta \log \pi(S_n, A_n) \sum_{m=0}^N r(S_m, A_m) \right], \tag{74}$$

which implies (22). This can be seen by defining the $n$-step transition function $p_n^\pi(s', s, a)$, in analogy to (53), via

$$\int_\Lambda p_n^\pi(s', s, a) \mathrm{d}s' = \mathbb{P}_\pi (S_n \in \Lambda | S_0 = s, A_0 = a), \tag{75}$$

and noting that for $m < n$ it holds

$$\mathbb{E}[\nabla_\theta \log \pi(S_n, A_n) r(S_m, A_m)] \tag{76a}$$

$$= \int_\mathcal{S} \int_\mathcal{A} \int_\mathcal{S} \int_\mathcal{A} \rho_m(s_m) \pi(s_m, a_m) r(s_m, a_m) p_{n-m}^\pi(s_n, s_m, a_m) \nabla_\theta \log \pi(s_n, a_n) \pi(s_n, a_n) \mathrm{d}s_m \mathrm{d}a_m \mathrm{d}s_n \mathrm{d}a_n \tag{76b}$$

$$= \int_\mathcal{S} \int_\mathcal{A} \int_\mathcal{S} \rho_m(s_m) \pi(s_m, a_m) r(s_m, a_m) p_{n-m}^\pi(s_n, s_m, a_m) \left( \int_\mathcal{A} \nabla_\theta \log \pi(s_n, a_n) \pi(s_n, a_n) \mathrm{d}a_n \right) \mathrm{d}s_m \mathrm{d}a_m \mathrm{d}s_n \tag{76c}$$

$$= 0,$$

---

[10]Note that Proposition 2.4 holds for any arbitrary function $r \colon \mathcal{S} \times \mathcal{A} \to \mathbb{R}$.

since

$$\int_{\mathcal{A}} \nabla_\theta \log \pi(s_n, a_n) \pi(s_n, a_n) \mathrm{d}a_n = \nabla_\theta \int_{\mathcal{A}} \pi(s_n, a_n) \mathrm{d}a_n = 0. \tag{77}$$

We can readily prove (24) in the state-space perspective by noting that

$$\nabla_\theta J(\pi_\theta) = \int_{\mathcal{S}} \eta^\pi(s) \int_{\mathcal{A}} \nabla_\theta \pi_\theta(s, a) Q^\pi(s, a) \mathrm{d}s \, \mathrm{d}a = \int_{\mathcal{S}} \eta^\pi(s) \int_{\mathcal{A}} \nabla_\theta \pi_\theta(s, a) (Q^\pi(s, a) - b(s)) \mathrm{d}s \, \mathrm{d}a \tag{78}$$

since

$$\int_{\mathcal{A}} b(s) \nabla_\theta \pi_\theta(s, a) \mathrm{d}a = b(s) \nabla_\theta \int_{\mathcal{A}} \pi_\theta(s, a) \mathrm{d}a = 0. \tag{79}$$

Then, by choosing $r(s, a) = \nabla_\theta \log \pi_\theta(s, a)(Q^\pi(s, a) - V^\pi(s))$ in Proposition 2.4, we arrive at (24) in the trajectory-based perspective. $\qquad\square$

**Lemma C.3** (Unrolling for deterministic policies). *For any $l \geq 1$ it holds that*

$$\nabla_\theta V^\mu(s) = \sum_{n=0}^{l-1} \int_{\mathcal{S}} p_n^\mu(s', s) \nabla_\theta \mu_\theta(s')^\top \nabla_a Q^\mu(s', a) \Big|_{a = \mu_\theta(s')} \mathrm{d}s' + \int_{\mathcal{S}} p_l^\mu(s', s) \nabla_\theta V^\mu(s') \mathrm{d}s'. \tag{80}$$

*Proof.* The proof of the unrolling lemma for deterministic policies follows the same idea as for stochastic policies stated in Lemma C.2. In analogy to (53), let us define the $n$-step transition function $p_n^\mu(s', s)$ via

$$\int_\Lambda p_n^\mu(s', s) \mathrm{d}s' = \mathbb{P}_\mu \left( S_n \in \Lambda | S_0 = s \right). \tag{81}$$

As before, let us first consider $l = 1$. Recalling the Bellman equation, $V^\mu(s) = r(s, \mu(s)) + \mathbb{E}_\mu \left[ V^\mu(S_1) | S_0 = s \right]$, stated already in (45), we can compute

$$\nabla_\theta V^\mu(s) = \nabla_\theta \left( r(s, \mu_\theta(s)) + \int_{\mathcal{S}} p(s', s, \mu_\theta(s)) V^\mu(s') \mathrm{d}s' \right) \tag{82a}$$

$$= \nabla_\theta \mu_\theta(s)^\top \nabla_a r(s, a) \Big|_{a = \mu_\theta(s)}$$
$$+ \int_{\mathcal{S}} \left( p(s', s, \mu_\theta(s)) \nabla_\theta V^\mu(s') + \nabla_\theta \mu_\theta(s)^\top \nabla_a p(s', s, a) \Big|_{a = \mu_\theta(s)} V^\mu(s') \right) \mathrm{d}s' \tag{82b}$$

$$= \nabla_\theta \mu_\theta(s)^\top \nabla_a \left( r(s, a) + \int_{\mathcal{S}} p(s', s, a) V^\mu(s') \mathrm{d}s' \right) \Big|_{a = \mu_\theta(s)}$$
$$+ \int_{\mathcal{S}} p(s', s, \mu_\theta(s)) \nabla_\theta V^\mu(s') \mathrm{d}s' \tag{82c}$$

$$= \nabla_\theta \mu_\theta(s)^\top \nabla_a Q^\mu(s, a) \Big|_{a = \mu_\theta(s)} + \int_{\mathcal{S}} p_1^\mu(s', s) \nabla_\theta V^\mu(s') \mathrm{d}s'. \tag{82d}$$

For notational convenience, let us define

$$\varphi_\theta(s) := \nabla_\theta \mu_\theta(s)^\top \nabla_a Q^\mu(s, a) \Big|_{a = \mu_\theta(s)}. \tag{83}$$

For the induction step, we can then compute

$$\nabla_\theta V^\mu(s) = \sum_{n=0}^{l-2} \int_{\mathcal{S}} p_n^\mu(s', s) \varphi_\theta(s') \mathrm{d}s' + \int_{\mathcal{S}} p_{l-1}^\mu(s', s) \left( \varphi_\theta(s') + \int_{\mathcal{S}} p_1^\mu(s'', s') \nabla_\theta V^\mu(s'') \mathrm{d}s'' \right) \mathrm{d}s' \tag{84a}$$

$$= \sum_{n=0}^{l-1} \int_{\mathcal{S}} p_n^\mu(s', s) \varphi_\theta(s') \mathrm{d}s' + \int_{\mathcal{S}} \int_{\mathcal{S}} p_{l-1}^\mu(s', s) p_1^\mu(s'', s') \nabla_\theta V^\mu(s'') \mathrm{d}s'' \mathrm{d}s' \tag{84b}$$

$$= \sum_{n=0}^{l-1} \int_{\mathcal{S}} p_n^\mu(s', s) \varphi_\theta(s') \mathrm{d}s' + \int_{\mathcal{S}} p_l^\mu(s', s) \nabla_\theta V^\mu(s') \mathrm{d}s', \tag{84c}$$

where we used (82) in the first line. $\qquad\square$

*Proof of Proposition 2.9.* The proof is analog to the one of Proposition 2.6. First, we see that in the limit of $l \to \infty$, the second summand of (80) vanishes. We can then compute

$$\nabla_\theta J(\mu_\theta) = \int_\mathcal{S} \rho_0(s_0) \nabla_\theta V^\mu(s_0) \mathrm{d}s_0 \tag{85a}$$

$$= \int_\mathcal{S} \rho_0(s_0) \left( \sum_{n=0}^\infty \int_\mathcal{S} p_n^\mu(s, s_0) \nabla_\theta \mu_\theta(s)^\top \nabla_a Q^\mu(s, a)\Big|_{a=\mu_\theta(s)} \mathrm{d}s \right) \mathrm{d}s_0 \tag{85b}$$

$$= \int_\mathcal{S} \eta^\mu(s) \nabla_\theta \mu_\theta(s)^\top \nabla_a Q^\mu(s, a)\Big|_{a=\mu_\theta(s)} \mathrm{d}s \tag{85c}$$

$$= \mathbb{E}_\mu[N+1] \, \mathbb{E}_{s \sim \rho^\mu} \left[ \nabla_\theta \mu_\theta(s)^\top \nabla_a Q^\mu(s, a)\Big|_{a=\mu_\theta(s)} \right], \tag{85d}$$

where we have used

$$\eta^\mu = \rho^\mu \int_\mathcal{S} \eta^\mu(s) \mathrm{d}s = \rho^\mu \, \mathbb{E}_\mu[N+1] \tag{86}$$

via definition (8) and Lemma 2.3 (however, replacing the stochastic policy $\pi$ with the deterministic policy $\mu$). Finally, by choosing $r(s, a) = \nabla_\theta \mu_\theta(s)^\top \nabla_a Q^\mu(s, a)_{|a=\mu_\theta(s)}$ in the deterministic policy version of Proposition 2.4 and using identity (85d), we readily get

$$\mathbb{E}_\mu \left[ \sum_{n=0}^N \nabla_\theta \mu_\theta(S_n)^\top \nabla_a Q^{\mu_\theta}(S_n, a)\Big|_{a=\mu_\theta(S_n)} \right]. \tag{87}$$

$\square$

*Proof of Corollary 2.10.* Using the Bellman equation (46), we can compute

$$\nabla_a Q^\mu(s, a) = \nabla_a r(s, a) + \int_\mathcal{S} \nabla_a p(s', s, a) V^\mu(s') \mathrm{d}s' \tag{88a}$$

$$= \nabla_a r(s, a) + \int_\mathcal{S} \nabla_a \log p(s', s, a) p(s', s, a) V^\mu(s') \mathrm{d}s' \tag{88b}$$

$$= \nabla_a r(s, a) + \mathbb{E}_{s' \sim p(\cdot, s, a)} \left[ \nabla_a \log p(s', s, a) V^\mu(s') \right]. \tag{88c}$$

Using Proposition 2.9, we get

$$\nabla_\theta J(\mu_\theta) = \mathbb{E}_\mu \left[ \sum_{n=0}^N \nabla_\theta \mu_\theta(S_n)^\top \nabla_a Q^{\mu_\theta}(S_n, a)\Big|_{a=\mu_\theta(S_n)} \right] \tag{89a}$$

$$= \mathbb{E}_\mu \left[ \sum_{n=0}^N \nabla_\theta \mu_\theta(S_n)^\top \left( \nabla_a r(S_n, a) + \mathbb{E}_{s' \sim p(\cdot, S_n, a)} \left[ \nabla_a \log p(s', S_n, a) V^\mu(s') \right] \right) \Big|_{a=\mu_\theta(S_n)} \right] \tag{89b}$$

$$= \mathbb{E}_\mu \left[ \sum_{n=0}^N \nabla_\theta \mu_\theta(S_n)^\top \left( \nabla_a r(S_n, a) + \mathbb{E}_{\mu_\theta} \left[ \nabla_a \log p(S_{n+1}, S_n, a) V^\mu(S_{n+1}) \right] \right) \Big|_{a=\mu_\theta(S_n)} \right] \tag{89c}$$

$$= \mathbb{E}_\mu \left[ \sum_{n=0}^N \nabla_\theta \mu_\theta(S_n)^\top \left( \nabla_a r(S_n, a) + \nabla_a \log p(S_{n+1}, S_n, a) V^\mu(S_{n+1}) \right) \Big|_{a=\mu_\theta(S_n)} \right], \tag{89d}$$

where we used the tower property in the last line. The second equality follows from plugging in (88) into the second equation of Proposition 2.9. $\square$

*Remark* C.4 (Connection to stochastic optimal control gradient). From Corollary 2.10 we can recover the gradient estimator of stochastic optimal control problems with random stopping times, as for instance stated in Corollary 3.3 in Ribera Borrell et al. (2024). To this end, we realize that the continuous time SDE

$$\mathrm{d}X_s^\mu = (b + \sigma\mu)(X_s^\mu) \, \mathrm{d}t + \sigma(X_s^\mu) \, \mathrm{d}W_s, \tag{90}$$

where $W$ is standard Brownian motion, $b : \mathbb{R}^d \to \mathbb{R}^d$ an arbitrary drift and $\sigma : \mathbb{R}^d \to \mathbb{R}^{d \times d}$ is the diffusion matrix, can be discretized via the Euler-Maruyama scheme

$$S_{n+1} = S_n + (b + \sigma\mu)(S_n)\Delta t + \sigma(S_n)\xi_{n+1}\sqrt{\Delta t}, \tag{91}$$

with step size $\Delta t > 0$ and Gaussian increment $\xi_{n+1} \sim \mathcal{N}(0, \mathrm{Id})$. Calling the terminal time $T = N\Delta t$, the continuous control costs

$$J(\mu) = \mathbb{E}\left[\int_0^T \left(f + \frac{1}{2}\|\mu\|^2\right)(X_s^\mu)\,\mathrm{d}t + g(X_T^\mu)\right] \tag{92}$$

correspond to $r(s, a) = -\left(f(s) + \frac{1}{2}\|a\|^2\right)\Delta t\,\mathbb{1}_{\mathcal{T}^c}(s) - g(s)\,\mathbb{1}_{\mathcal{T}}(s)$ in the discrete setting (5) and we can thus compute

$$\nabla_a r(s, a) = -a\Delta t\,\mathbb{1}_{\mathcal{T}^c}(s), \tag{93}$$

$$\nabla_a \log p(S_{n+1}, S_n, a) = \sigma^{-1}(S_n)\left(S_{n+1} - (S_n + (b + \sigma a)(S_n)\Delta t)\right) = \xi_{n+1}\sqrt{\Delta t}. \tag{94}$$

We can therefore see that (27) from Corollary 2.10 corresponds to the time-discretized version of Corollary 3.3 in Ribera Borrell et al. (2024). We highlight that a derivation in discrete time including random stopping times has not been rigorously done before (cf. Hartmann & Schütte (2012), where it has been conjectured that the formula is inexact due to the random time).

**Corollary C.5** (Alternative trajectory-based versions of the model-based policy gradient for deterministic policies). *For the gradient of the expected return* (5) *it holds*

$$\nabla_\theta J(\mu_\theta) = \mathbb{E}_\mu\left[\sum_{n=0}^N \nabla_\theta \mu_\theta(S_n)^\top\left(\nabla_a r(S_n, a) + \sum_{m=0}^N r(S_m, \mu_\theta(S_m))\nabla_a \log p(S_{n+1}, S_n, a)\right)\Big|_{a=\mu_\theta(S_n)}\right] \tag{95}$$

$$= \mathbb{E}_\mu\left[\sum_{n=0}^N \nabla_\theta \mu_\theta(S_n)^\top\left(\nabla_a r(S_n, a) + \sum_{m=n+1}^N r(S_m, \mu_\theta(S_m))\nabla_a \log p(S_{n+1}, S_n, a)\right)\Big|_{a=\mu_\theta(S_n)}\right] \tag{96}$$

$$= \mathbb{E}_\mu\left[\sum_{n=0}^N \nabla_\theta \mu_\theta(S_n)^\top\nabla_a\left(Q^{\mu_\theta}(S_n, a)\Big|_{a=\mu_\theta(S_n)} - b(S_n)\right)\right]. \tag{97}$$

*where* $b : \mathcal{S} \to \mathbb{R}$ *is an arbitrary function (sometimes called* baseline*).*

*Proof of Corollary C.5.* The proof of the alternative versions of the deterministic policy gradient follows the same idea as for stochastic policies stated in Corollary 2.8.

We first note that for an arbitrary function $\varphi : \mathcal{S}^{n+2} \to \mathbb{R}$ it holds

$$\mathbb{E}_\mu\left[\varphi(S_0, \dots, S_{n+1})V^\mu(S_{n+1})\right] = \mathbb{E}_\mu\left[\varphi(S_0, \dots, S_{n+1})\mathbb{E}_\mu\left[\sum_{m=n+1}^N r(S_m, \mu(S_m))\Big| S_n, A_n\right]\right] \tag{98a}$$

$$= \mathbb{E}_\mu\left[\mathbb{E}_\mu\left[\varphi(S_0, \dots, S_{n+1})\sum_{m=n+1}^N r(S_m, \mu(S_m))\right]\right] \tag{98b}$$

$$= \mathbb{E}_\mu\left[\varphi(S_0, \dots, S_{n+1})\sum_{m=n+1}^N r(S_m, \mu(S_m))\right], \tag{98c}$$

where we used the tower property, the fact that $\varphi(S_0, \dots, S_{n+1})$ is $\mathcal{F}_{n+1}$-measurable and the Markov property. Equality (96) now follows from choosing $\varphi(S_0, \dots, S_{n+1}) = \nabla_\theta \mu_\theta(S_n)^\top\nabla_a \log p(S_{n+1}, S_n, a))\Big|_{a=\mu_\theta(S_n)}$. For this choice we can even show that

$$\mathbb{E}_\mu\left[\nabla_\theta \mu_\theta(S_n)^\top V^\pi(S_{n+1})\nabla_a \log p(S_{n+1}, S_n, a)\big|_{a=\mu_\theta(S_n)}\right]$$
$$= \mathbb{E}_\mu\left[\nabla_\theta \mu_\theta(S_n)^\top\sum_{m=0}^N r(S_m, \mu_\theta(S_m))\nabla_a \log p(S_{n+1}, S_n, a)\big|_{a=\mu_\theta(S_n)}\right], \tag{99}$$

which implies (95). This can be seen by using the $n$-step transition function $p_n^\mu(s', s, \mu(s))$, which is stated in Corollary 2.8 for stochastic policies, here denoted by $p_n^\mu(s', s)$, as well as by noting that for $m < n + 1$ it holds

$$\mathbb{E}_\mu \left[ \nabla_\theta \mu_\theta(S_n)^\top r(S_m, \mu_\theta(S_m)) \nabla_a \log p(S_{n+1}, S_n, a) \big|_{a=\mu_\theta(S_n)} \right] \tag{100a}$$

$$= \int_\mathcal{S} \int_\mathcal{S} \int_\mathcal{S} \rho_m(s_m) r(s_m, \mu(s_m)) p_{n-m}^\mu(s_n, s_m) \nabla_\theta \mu(s_n)^\top p_1^\mu(s_{n+1}, s_n) \nabla_a \log p(s_{n+1}, s_n, a) \big|_{a=\mu(s_n)} \mathrm{d}s_m \mathrm{d}s_n \mathrm{d}s_{n+1} \tag{100b}$$

$$= \int_\mathcal{S} \int_\mathcal{S} \rho_m(s_m) r(s_m, \mu(s_m)) p_{n-m}^\mu(s_n, s_m) \nabla_\theta \mu(s_n)^\top \left( \int_\mathcal{S} p_1^\mu(s_{n+1}, s_n) \nabla_a \log p(s_{n+1}, s_n, a) \big|_{a=\mu(s_n)} \mathrm{d}s_{n+1} \right) \mathrm{d}s_m \mathrm{d}s_n \tag{100c}$$

$$= 0,$$

since for any tuple $(s', s, a) \in \mathcal{S} \times \mathcal{S} \times \mathcal{A}$ it holds that

$$\int_\mathcal{S} \nabla_a \log p(s', s, a) p(s', s, a) \mathrm{d}s' = \int_\mathcal{S} \nabla_a p(s', s, a) \mathrm{d}s' = \nabla_a \int_\mathcal{S} p(s', s, a) \mathrm{d}s' = 0. \tag{101}$$

We can readily prove (97) in the state-space perspective by noting that

$$\nabla_\theta J(\mu_\theta) = \int_\mathcal{S} \eta^\mu(s) \nabla_\theta \mu_\theta(s)^\top \nabla_a Q^\pi(s, a) \big|_{a=\mu_\theta(s)} \mathrm{d}s = \int_\mathcal{S} \eta^\mu(s) \nabla_\theta \mu_\theta(s)^\top \nabla_a (Q^\pi(s, a) - b(s)) \big|_{a=\mu_\theta(s)} \mathrm{d}s. \tag{102}$$

By choosing $r(s, a) = r(s, \mu_\theta(s)) = \nabla_\theta \mu_\theta(s) \nabla_a (Q^\mu(s, a) - b(s)) \big|_{a=\mu_\theta(s)}$ in Proposition 2.4 we arrive at (97) in the trajectory-based perspective. $\qquad\qquad\square$

## D. Computational details

In this section we provide computational details that are necessary for the numerical approximation of the gradients discussed in the main part.

For policy gradients of stochastic policies we consider non-actor-critic approaches where the $Q$-value function is estimated, but not learned. The alternative objectives corresponding to the trajectory PG stated in (22) and to the state-space PG stated in (16) are given by

$$J_{\mathrm{eff}}^{\mathrm{traj}}(\pi_\theta, \pi_\vartheta) := \mathbb{E}_{\pi_\vartheta} \left[ \sum_{n=0}^N \log \pi_\theta(S_n, A_n) \sum_{m=0}^N r(S_m, A_m) \right], \tag{103}$$

$$J_{\mathrm{eff}}^{\mathrm{state}}(\pi_\theta, \pi_\vartheta) := \mathbb{E}_{\pi_\vartheta} [N + 1] \mathbb{E}_{\substack{s \sim \rho^{\pi_\vartheta} \\ a \sim \pi_\vartheta(s, \cdot)}} \left[ \log \pi_\theta(s, a) Q^{\pi_\vartheta}(s, a) \right]. \tag{104}$$

We refer to Algorithms 1 and 2 for implementational details.

For policy gradients of deterministic policies we consider non-actor-critic approaches as well. Recall that this is possible due to the model-based formulas provided in (27) and (95). Moreover, we have assumed that the state-action probability density is Gaussian and is motivated by a controlled SDE (see Remark C.4). The corresponding alternative objectives are given by

$$J_{\mathrm{eff}}^{\mathrm{traj}}(\mu_\theta, \mu_\vartheta) := \mathbb{E}_{\mu_\vartheta} \left[ \sum_{n=0}^N \mu_\theta(S_n)^\top \left( \nabla_a r(S_n, a) \big|_{a=\mu_\vartheta(S_n)} + \sum_{m=0}^N r(S_m, \mu_\vartheta(S_m)) \xi_{n+1} \sqrt{\Delta t} \right) \right], \tag{105}$$

$$J_{\mathrm{eff}}^{\mathrm{state}}(\mu_\theta, \mu_\vartheta) := \mathbb{E}_{\mu_\vartheta} [N + 1] \mathbb{E}_{\substack{s \sim \rho^{\mu_\vartheta}, \\ s' \sim p^{\mu_\vartheta}(\cdot, s)}} \left[ \mu_\theta(s)^\top \left( \nabla_a r(s, a) \big|_{a=\mu_\vartheta(s)} + V^{\mu_\vartheta}(s') \xi_{n+1} \sqrt{\Delta t} \right) \right]. \tag{106}$$

We refer to Algorithms 3 and 4 for implementational details and note that algorithm Algorithm 3 is already sketched in Quer & Ribera Borrell (2024).

## E. Experimental details and additional experiments

In this section we provide further details on the numerical examples presented in Section 3.

---

**Algorithm 1** Trajectory Policy Gradient (REINFORCE with random time horizon).

---

1: Initialize stochastic policy $\pi_\theta$ with random parameters.
2: Choose a gradient based optimization algorithm, a learning rate $\lambda$, a sample size $K$, and a stopping criterion.
3: **repeat**
4:  Simulate $K$ samples of trajectories following the policy $\pi_\theta$, where the $k$-th trajectory has runtime $N^{(k)}$.
5:  Estimate alternative objective stated in (103) and compute the gradient via automatic differentiation

$$\widehat{J}_{\text{eff}}^{\text{traj}} = \frac{1}{K} \sum_{k=1}^{K} \sum_{n=0}^{N^{(k)}} \log \pi_\theta(S_n^{(k)}, A_n^{(k)}) \sum_{m=0}^{N^{(k)}} r(S_m^{(k)}, A_m^{(k)}).$$

6:  Update policy network parameters based on the optimization algorithm.
7: **until** stopping criterion is fulfilled.

---

---

**Algorithm 2** State-space Policy Gradient.

---

1: Initialize stochastic policy $\pi_\theta$ with random parameters.
2: Choose a gradient based optimization algorithm, a learning rate $\lambda$, a sample size $K$ for the trajectories, a sample size $M$ for the experiences and a stopping criterion.
3: **repeat**
4:  Simulate $K$ samples of trajectories following the policy $\pi_\theta$, where the $k$-th trajectory has runtime $N^{(k)}$.
5:  Compute estimate $\widehat{Z}^\pi = \frac{1}{K} \sum_{k=1}^{K} N^{(k)} + 1$.
6:  **for** each time step $n$ **do**
7:    Compute return $G_n^{(k)} = \sum_{m=n}^{N^{(k)}} r(S_m^{(k)}, A_m^{(k)})$ and store the tuple $(S_n^{(k)}, A_n^{(k)}, G_n^{(k)})$ in memory.
8:  Sample $M$ experiences from memory $\{\widetilde{S}^{(m)}, \widetilde{A}^{(m)}, \widetilde{G}^{(m)}\}_{m=1}^{M}$, where $\widetilde{S}^{(m)}, \widetilde{A}^{(m)}$ are unbiased samples from $\rho^\pi$ and $\pi(\widetilde{S}^{(m)}, \cdot)$, respectively, and $\widetilde{G}^{(m)}$ estimates $Q^\pi(\widetilde{S}^{(m)}, \widetilde{A}^{(m)})$.
9:  Estimate alternative objective stated in (104) and compute the gradient via automatic differentiation

$$\widehat{J}_{\text{eff}}^{\text{state}} = \widehat{Z}^\pi \frac{1}{M} \sum_{m=1}^{M} \log \pi_\theta(\widetilde{S}^{(m)}, \widetilde{A}^{(m)}) \widetilde{G}^{(m)}.$$

10:  Update policy network parameters based on the optimization algorithm.
11:  Empty memory.
12: **until** stopping criterion is fulfilled.

---

---

**Algorithm 3** Trajectory and model-based Deterministic Policy Gradient.

---

1: Initialize deterministic policy $\mu_\theta$ with random parameters.
2: Choose a gradient based optimization algorithm, a learning rate $\lambda$, a sample size $K$, and a stopping criterion.
3: **repeat**
4:  Simulate $K$ samples of trajectories following the policy $\mu_\theta$.
5:  Estimate alternative objective stated in (105) and compute the gradient via automatic differentiation

$$\widehat{J}_{\text{eff}}^{\text{traj}} = \frac{1}{K} \sum_{k=1}^{K} \left( \sum_{n=0}^{N^{(k)}} \mu_\theta(S_n^{(k)})^\top \left( \nabla_a r(S_n^{(k)}, a)\big|_{a=\mu_\vartheta(S_n^{(k)})} + \left( \sum_{m=0}^{N^{(k)}} r(S_m^{(k)}, \mu_\vartheta(S_m^{(k)})) \right) \xi_{n+1}^{(k)} \sqrt{\Delta t} \right) \right).$$

6:  Update the parameters $\theta$ based on the optimization algorithm.
7: **until** stopping criterion is fulfilled.

---

---

**Algorithm 4** State-space and model-based Deterministic Policy Gradient.

---

1: Initialize deterministic policy $\mu_\theta$ with random parameters.
2: Choose a gradient based optimization algorithm, a learning rate $\lambda$, a sample size $K$ for the trajectories, a sample size $M$ for the experiences and a stopping criterion.
3: **repeat**
4:     Simulate $K$ trajectories following the policy $\mu_\theta$ and store them in memory.
5:     Compute estimate $\widehat{Z}^\mu = \frac{1}{K} \sum_{k=1}^{K} N^{(k)} + 1$.
6:     **for** each time step $n$ **do**
7:         Compute return $G_{n+1}^{(k)} = \sum_{m=n+1}^{N} r(S_m^{(k)}, \mu_\theta(S_m^{(k)}))$ and store the tuple $(S_n^{(k)}, S_{n+1}^{(k)}, \xi_{n+1}^{(k)}, G_{n+1}^{(k)})$ in memory.
8:     Sample $M$ experiences from memory $\{\widetilde{S}^{(m)}, \widetilde{S}'^{(m)} \widetilde{\xi}^{(m)}, \widetilde{G}^{(m)}\}_{m=1}^{M}$ where $\widetilde{S}^{(m)}$ is an unbiased sample from $\rho^\mu$, $\widetilde{\xi}^{(m)} \sim \mathcal{N}(0, \mathrm{Id})$, and $\widetilde{G}^{(m)}$ estimates $V^\mu(\widetilde{S}'^{(m)})$.
9:     Estimate alternative objective $J_{\mathrm{eff}}$ given in (106) and compute the gradient via automatic differentiation

$$\widehat{J}_{\mathrm{eff}}^{\mathrm{state}} = \widehat{Z}^\mu \frac{1}{M} \sum_{m=1}^{M} \mu_\theta(\widetilde{S}^{(m)})^\top \left( \nabla_a r(\widetilde{S}^{(m)}, a)\big|_{a=\mu_\vartheta(\widetilde{S}^{(m)})} + \widetilde{G}^{(m)} \widetilde{\xi}^{(m)} \sqrt{\Delta t} \right)$$

10:     Update policy network parameters based on the optimization algorithm.
11:     Empty memory.
12: **until** stopping criterion is fulfilled.

---

### E.1. Architecture of the neural networks

Let $d_{\mathrm{in}}, d_{\mathrm{out}} \in \mathbb{N}^+$ be the input and output dimensions of the feed-forward network $\varphi_\theta \colon \mathbb{R}^{d_{\mathrm{in}}} \to \mathbb{R}^{d_{\mathrm{out}}}$ defined by

$$\varphi_\theta(x) = \rho_{\mathrm{out}}(A_L \rho(A_{L-1} \rho(\cdots \rho(A_1 x + b_1) \cdots) + b_{L-1}) + b_L), \tag{107}$$

where $L$ is the number of layers $d_0 = d_{\mathrm{in}}, d_L = d_{\mathrm{out}}, A_l \in \mathbb{R}^{d_l \times d_{l-1}}$ and $b_l \in \mathbb{R}^{d_l}, 1 \le l \le L$ are the weights and the bias vectors for each layer and $\rho, \rho_{\mathrm{out}} \colon \mathbb{R} \to \mathbb{R}$ are the inner and outer nonlinear activation functions applied componentwise. The collection of matrices $A_l$ and vectors $b_l$ contains the learnable parameters $\theta \in \mathbb{R}^p$. For all the experiments we choose the inner and outer activation functions $\rho = \tanh$ and $\rho_{\mathrm{out}} = \mathrm{Id}$ if not otherwise stated.

For the stochastic policy experiments described in Sections 3.1 and 3.2 we consider a Gaussian stochastic policy $\pi(s, \cdot) = \mathcal{N}(\mu(s), \sigma^2(s)\,\mathrm{Id})$, for which $\mu$ and $\sigma$ are learnable functions represented by two $L = 3$ layer feed-forward neural networks sharing the first $L = 2$ layers (a so-called two-head neural network). To guarantee that $\sigma^2\,\mathrm{Id}$ is positive definite, the final activation function of the corresponding neural network is chosen as $\rho_{\mathrm{out}}^\sigma = x + \sqrt{x^2 + 1}$.

To ensure that the initial output of the networks is close to zero, the final layer weights and biases are initialized by sampling from the uniform distribution $\mathcal{U}(-5 \times 10^{-3}, 5 \times 10^{-3})$. We also note that each experiment requires only one CPU core, and the maximum value of allocated memory is set to 64 GB.

### E.2. Modified continuous mountain car problem

For the experiment described in Section 3.1 we consider a Gaussian stochastic policy for which $\mu$ and $\sigma$ are represented by a two-head neural network (see details in Appendix E.1) with $L = 3$ layers and $d_1 = d_2 = 32$ units. We compare the three different policy gradient formulas by implementing Algorithm 1 (*trajectory PG*), Algorithm 2 (*state-space PG*) and Algorithm 2 without estimating the $Z^\pi$-factor (*state-space PG unbiased*) for a batch of $K = 100$ trajectories, a batch of experiences containing all the information in the memory ($M = 100\%$ of the memory size), and we stop the optimization algorithm after $I = 5 \times 10^4$ gradient iterations. The best performing learning rates for each gradient approach are $\lambda_{\mathrm{traj}} = \lambda_{\mathrm{state}} = 10^{-4}$, $\lambda_{\mathrm{state}}^{\mathrm{biased}} = 5 \times 10^{-2}$, respectively. Note that for $\lambda_{\mathrm{state}}^{\mathrm{biased}} \ge 10^{-1}$ the optimization algorithm fails due to high instabilities which lead to long trajectories and hence exceed the allowed allocated memory.

### E.3. Two-joint robot arm (reacher)

For the experiment described in Section 3.2 we consider a Gaussian stochastic policy with $L = 3$ layers and $d_1 = d_2 = 32$ units and compare the performance of Algorithm 1 (*trajectory PG*), Algorithm 2 (*state-space PG*), and Algorithm 2 without

estimating the $Z^\pi$-factor (*state-space PG unbiased*) for $K = 100$ trajectories, $M = 100\%$ of experiences in memory, and $I = 10^4$ gradient iterations. The best performing learning rates for each gradient approach are $\lambda_{\text{traj}} = \lambda_{\text{state}} = 5 \times 10^{-4}$, $\lambda_{\text{state}}^{\text{biased}} = 5 \times 10^{-2}$, respectively.

### E.4. Importance sampling of hitting times in molecular dynamics

For the experiment described in Section 3.3 we consider a deterministic policy represented by a neural network with $L = 2$ layers and $d_1 = 32$. We compare the three different model-based policy gradient formulas for deterministic policies by implementing Algorithm 3 (*trajectory DPG*), Algorithm 4 (*state-space DPG*) and Algorithm 4 without estimating the $Z^\mu$-factor (*state-space DPG unbiased*) for $K = 500$ trajectories, $M = 100\%$ of experiences in memory, and $I = 5 \times 10^4$ gradient iterations. The best performing learning rates for each gradient approach are $\lambda_{\text{traj}} = \lambda_{\text{state}} = 2 \times 10^{-3}$, $\lambda_{\text{state}}^{\text{biased}} = 5 \times 10^{-1}$, respectively.

