# OpenReview forum: "Reinforcement Learning with Random Time Horizons"
_ICML.cc/2025/Conference — ICML 2025 poster_

### Official Review · Reviewer_WigC · 2025-03-11

**Overall Recommendation:** 2

**Summary:**

The paper derives the policy gradient theorem for the setting where the MDP horizon is random (and typically policy dependent). Algorithmically, the "corrected" PG boils down to the standard PG with a multiplicative factor correction for the expected horizon length.

Numerical experiments are carried in two environments (continuous mountain car, reacher, hitting times in molecular
dynamics), demonstrating a certain advantage of the corrected PG computation (to the extent permitted by these experiments).

## update after rebuttal
I appreciate the rebuttal by the authors, however my main concern remains that the scope of this work sums up to a relatively straightforward extension to the policy gradient theorem. As a result I choose to maintain my initial rating.

**Claims And Evidence:**

The theoretical claims are, the experiments are quite minimal.

They show an advantage of corrected PG in a very limited experimental setup (just three environments, arbitrarily fixed number of iterations, etc.). I didn't find how many seeds were used in the experiments, standard errors of the results, etc.

In any case, I believe the corrected PG expression has merit regardless of the experiments. However, if it does not make a great difference in practice, this should also be discussed and it is currently unclear due to the minimal experimental setup.

**Essential References Not Discussed:**

I think citing the Agarwal et el. 2019 (RL Theory Book) could provide some additional context on basic matters (for example, the state space perspective is prevalent in the theory of RL, these are called occupancy measures).

Agarwal, A., Jiang, N., Kakade, S. M., & Sun, W. (2019). Reinforcement learning: Theory and algorithms.

**Experimental Designs Or Analyses:**

I looked at both experiments, didn't find any particular issues, except for the limited setup.

**Methods And Evaluation Criteria:**

The theoretical arguments are clear and make sense. The experimental setup is limited.

**Other Comments Or Suggestions:**

None

**Other Strengths And Weaknesses:**

### Strengths

* The paper points to the fact that vanilla policy gradients are "incorrect" for random time horizons in the sense that there is a normalization factor missing from their expressions, that is non constant over training. This means that the effective step size used by a vanilla PG algorithm changes over the course of training, which is generally undesirable when not intended.

### Weaknesses

* The technical contribution here is quite minimal in terms of RL theory, and the experiments are very scarce. In particular, vanilla PGs are rarely used in practice. I would be more curious to see if this observation has an effect on more modern policy optimization algorithms such as PPO, and in more challenging environments.

* The paper goes through a rather elaborate exposition of well known facts; e.g., the state-space perspective (known as occupancy measures), Figure 1, and Lemma 2.3 (indeed, adapted to the random horizon setup, but still).


The bottom line is that while I feel random time horizons and their consequence on PGs and PO algorithms in general should be considered more carefully in applied RL, there isn't sufficient substance in this paper for publication as is.

**Questions For Authors:**

None

**Relation To Broader Scientific Literature:**

Yes, the relevant PG papers Sutton et al. 99 and Silver et al. 2014 are mentioned.

Other related works are mentioned in the introduction.

**Theoretical Claims:**

I went over proof of Proposition 2.6 and Lemma C.2, they look ok.

---

> ### Author Rebuttal · Authors · 2025-04-01
>
> Dear Reviewer WigC. We thank you very much for your careful review and are happy that you - similar to the other reviewers - think our "corrected" policy gradient in principle "has merit" and adds a meaningful contribution to the reinforcement learning community. We want to highlight that - compared to the standard policy gradient - the added multiplicative factor can indeed vary substantially over the course of the optimization, see the "effective learning rates" e.g. in Figures 2 - 4 and also Remark 2.7. This not only leads to small improvements, but to major speed-ups (time reductions of 60-80%) as well as to improved convergence in our experiments (see Section 3).
>
> Thank you for your comment regarding PPO - this is very valuable in order to better position our achievements in the context of the algorithms that are most commonly used in practice. We believe that our novel formulas can also be applied to PPO and other advanced algorithms, however, in the short time of the rebuttal phase, we could not run experiments yet. But let us make the theoretical agrument.
>
> PPO builds on TRPO, which aims to maximize the expected reward, while making sure that in each gradient step $ \mathbb{E} [\operatorname{KL}( \widetilde{\pi}(\cdot, s)  | \pi(\cdot, s)) ] \le \delta$ holds, $\widetilde{\pi}$ is the old policy, i.e. the policy is not changed too much. In practice, the constrained optimization is typically conducted with conjugate gradient algorithms and line searches, where a linear approximation of the expected reward and a quadratic approximation of the constrain is employed. This then results in
> $$\widetilde{G}^{(k)} := {(H^{-1})}^{(k)} G^{(k)},$$
>
> where $G^{(k)}$ is the policy gradient and $H^{(k)}$ is the Hessian of the estimated KL divergence. One then considers the update
>
> $$\theta^{(k+1)}=\theta^{(k)}+\alpha^{(k)} \sqrt{\frac{2 \delta}{\widetilde{G}^{(k)} \cdot H^{(k)} \widetilde{G}^{(k)}}} \widetilde{G}^{(k)},$$
>
> where a good value for $\alpha^{(k)} > 0$ can be found via line search. In the setting of random time horizons, the TRPO algorithm uses the "incorrect" gradient estimator $G^{(k)}$, however, the scaling and the "line searching" of the step size may cure potentially bad choices. One can readily replace $G^{(k)}$ with our novel formulas that give the "correct" gradient in random time horizon settings. We anticipate that this should further improve the performance of TRPO, however, due to the limited time in the rebuttal, we need to leave experiments for the next weeks. We will, however, incorporate them in the final version of the paper, thus being able to better contextualize the practical implications within contemporary RL research.
>
> PPO makes even further approximations/simplifications and considers the loss
> $$ \mathcal{L}(\pi) = \mathbb{E}\left[\min \left( \log \pi (a, s) A, \mathrm{clip}\left(\log \pi(a, s), 1-\varepsilon, 1 + \varepsilon \right) A \right) \right], $$
>
> where $A$ is the advantage function and $\varepsilon > 0$ is a hyperparameter. Also here, we can incoporate the findings of our novel gradient estimator, by adding the scaling factor $\mathbb{E}[N + 1]$ to "fix" the "incorrect" gradient estimators, cf. Propositions 2.4 and 2.6.
>
> Finally, we note that in Section 3 we compared the vanilla gradient estimators on purpose in order to properly study the effect of the random time horizons on the (novel) "correct" and (typically used) "incorrect" gradient formulas and in order to exclude confounding effects.
>
> Additional comments:
>
>  - Thank you for pointing out Agarwal et al., 2019, as a good reference. We will add it to the revised version upon acceptance.
>  - *"More challenging environments."* For the rebuttal, we ran the *Hopper* environment, see https://tinyurl.com/mtxcnudu. One can see similar advantages of our gradient estimator compared to the standard one.
>  - *"Arbitrarily fixed number of iterations."* We chose the number of iterations such that all algorithms converge.
>  - *"Didn't find how many seeds were used in the experiments, standard errors of the results."* The experiments were run for three different seeds. Notice that in the left plot of Fig. 2-4 the transparency values indicate different seeds. Observe consistent behavior, see, e.g., https://tinyurl.com/3r3y6bxm. We will add more elaborate plots (e.g. containing standard deviations) in the final version.
>  - We agree that we state some known facts in the introduction, however, we would like to keep this since our results and story line heavily depends on concepts such as occupancy measures. Also, we think that certain connections between random time horizons and existing approaches (e.g. Lemma 2.2) are not well known within the community. In case we will have space issues, we will however consider shortening the introduction - thanks for this helpful advice!
>
> If you have any further questions or concerns, please let us know. Otherwise we would be happy if you consider reevaluating our contribution. Thank you very much!

---

### Official Review · Reviewer_TpCt · 2025-03-12

**Overall Recommendation:** 4

**Summary:**

In this paper, the authors consider the problem of undiscounted, random horizon RL.  In this setting, a learner is attempting to optimize the cumulative reward of a policy interacting with an MDP such that the horizon, $N$, is a possibly random stopping time adapted to the filtration of the episode thus far.  The authors develop formalism involving such MDPs from both a trajectory and state-space based perspective.  The authors proceed to prove a number of basic results about this setting before focusing on computing policy gradients to aid in RL.  They then interpret the difference between the true policy gradient and the standard computation using reinforce but ignoring the random horizon as a rescaling of the learning rate.

The authors then conduct a number of empirical investigations of their gradient computation as compared to that which ignores the random horizon in a modified mountain car environment, a reacher environment, and a problem in molecular dynamics.  The authors demonstrate that their approach improves significantly on that which ignores the random horizon.


####

The authors answered my questions and I maintain my initial (positive) score.

**Claims And Evidence:**

Yes.

**Essential References Not Discussed:**

N/A.

**Experimental Designs Or Analyses:**

I did not spend as much time perusing the precise experimental setup in the supplementary section, but from my reading of the main body's discussion of the experiments, it seems reasonable to me.

**Methods And Evaluation Criteria:**

I think the benchmarks do make sense.  They are clearly chosen so as to introduce random stopping times in a natural way and all three are fairly standard proofs of concept in RL, especially the first two.

**Other Comments Or Suggestions:**

See above.

**Other Strengths And Weaknesses:**

I think the paper overall clearly presents an important correction to prior policy gradient methods in settings where the horizon can vary in policy-dependent ways.  One weakness is the emphasis on stationarity, which is necessary for the state-dependent perspective to be time-invariant.  I wonder how reasonable this is as an assumption and the extent to which it can be removed?  I also think that a more clear discussion of this assumption could be included as I only realized that this was essential in lines 130-131 from the parenthetical clause.

**Questions For Authors:**

See weaknesses.

**Relation To Broader Scientific Literature:**

There has been quite a lot of prior work on policy gradient methods in a number of directions and they are of course fundamental to modern empirical RL pipelines.  This paper observes that there is a gap in the prior work in that it does not rigorously address the question of how such methods should change when the horizon is random as opposed to deterministically finite or infinite.  This is an important gap in cases where the horizon can vary quite a bit because the horizon itself can depend on the policy.

**Theoretical Claims:**

I did check the theoretical claims and, while somewhat basic, they are correct.

---

> ### Author Rebuttal · Authors · 2025-04-01
>
> Dear Reviewer TpCt. Thank you very much for your educated review. We are happy that you conclude that our contribution is closing an important gap in reinforcement learning and we appreciate that you value our numerical benchmarks as appropriate.
>
>
>
> Thank you also for raising the aspect of time-independent densities, leading to stationarity, which you spotted very well. In our paper we consider expected cumulative rewards of the form
>
> $$ J(\pi) = \mathbb E_\pi \left[  \sum\limits_{n=0}^N r(S_n, A_n) \right], $$
>
> where $N$ is a random stopping time, i.e. $N = \min(n \in \mathbb{N}: S_n \in \mathcal{T})$, and where $\mathcal{T}$ is some target set. In this setting the stationarity assumption is natural and we will highlight this more in the revised version of the paper. We could, however, replace $N$ with $\min(N, N_\mathrm{max})$, where $N_\mathrm{max} \in \mathbb{N}$ is a fixed value. This would in fact lead to time-dependend state densities as well as policies that are explicitly time-dependent. Our theory should still go through, however, would be notationaly more challenging and slighly change proof details and formulas. For instance, the value function would also be time-dependent in this case. Since the time-dependent case is typically less prominent in applications, we decided to focus on the time-independent case in our paper. We will, however, comment on this subtle issue in the revised version of the paper and thank you again for the comment.
>
> (Also note that in time-continuous (stochastic) optimal control theory time-independent problems correspond to elliptic partial differential equations, whereas time-dependent problems correspond to parabolic partial differential equations, leading to slightly different assumptions and to slightly modified algorithms.)
>
> Please do not hesitate to ask further questions, we would be happy to answer them.

---

### Official Review · Reviewer_aYkS · 2025-03-13

**Overall Recommendation:** 3

**Summary:**

This paper considers a more realistic setting of reinforcement learning when the time horizon is random rather than fixed finite or infinite. The authors extend the RL to incorporate random time horizons and present the expected returns under the random time horizon from both trajectory-based and state space based perspectives.  The author also present the corresponding gradient descent theorems for both cases with rigorous theoretical proofs. Multiple numerical experiments are presented to show the importance of using the proposed GD strategy when the time horizon is not fixed in real applications.

**Claims And Evidence:**

Yes, this paper is generally well-supported by both theoretical analysis and empirical results. The theorems are presented with clear description and rigorous proofs in the appendix. In addition, the experiments presented with real world applications support the claimed statement.

**Essential References Not Discussed:**

No, I am not aware of any.

**Experimental Designs Or Analyses:**

Both experiment setups in the paper are relatively simple RL tasks. While these setups do demonstrate the idea, they may not fully capture the challenges of real-world applications with random time horizons.

The selected baseline line in the paper is the standard policy gradient algorithm. I think the similar approach can be applied to other algorithm directly. And comparison against more advance RL algorithm that have strategies for handling variable time horizons are needed to support the claims of the paper.

**Methods And Evaluation Criteria:**

Yes, the proposed method is to address the gap between the standard formulation and the real application. Under the random time horizon assumption, the proposed method incorporate the randomness into the gradient calculation makes sense to me.

**Other Comments Or Suggestions:**

No other comments.

**Other Strengths And Weaknesses:**

Overall, the paper presents theoretical analysis and shows promising empirical results. Its main strengths lie in its originality and theoretical rigor. However, it could be improved by providing more comprehensive empirical evaluations and clearer guidelines for practical implementation. Despite these limitations, the work represents a valuable contribution to the field of reinforcement learning, particularly in handling the important and often overlooked aspect of random time horizons.

**Questions For Authors:**

1.Could you elaborate on how your approach compares to modern RL algorithms (like PPO, TRPO) that already have mechanisms for handling variable-length episodes?
2.Have you tested the approach on more complex environments with highly variable time horizons? Results from such environments would help demonstrate the method's scalability and practical utility.

**Relation To Broader Scientific Literature:**

The paper makes clear theoretical advances but could better contextualize its practical implications within contemporary RL research.

**Theoretical Claims:**

Yes, I read most parts of the proofs provided in the appendix. They looks correct to me. The authors are thorough in their derivations and clearly state their assumptions.

---

> ### Author Rebuttal · Authors · 2025-04-01
>
> Thank you very much for your review. We are happy that you value our "clear theoretical advances", addressing "the gap between the standard formulation and the real application", thus leading to a "valuable contribution to the field of reinforcement learning".
>
> Thanks for asking for more comprehensive empirical evaluations. First, note that we have already chosen examples that exhibit highly variable time horizons. This can be seen by the "effective learning rates" in Figures 2 - 4, which directly correspond to the current expected stopping times, see equation (21). For the rebuttal, we furthermore ran the *Hopper* environment, see https://tinyurl.com/mtxcnudu. One can see similar advantages of our gradient estimator compared to the standard one. Note also that in all experiments we can achieve major speed-ups compared to the standard policy gradient (time reductions of 60-80%) as well as improved convergence (see Section 3).
>
> Your comment regarding clearer guidelines for practical implementation is a good one. First, we want to refer to Appendix D, where we already stated computational details and implementation guidelines - see in particular Algorithms 1 - 4. If space permits, we will move one of the algorithm environments to the main part. Further, note that Appendix E contains details on the conducted experiments, in particular stating the used hyperparameters. In the revised version, we will make both appendices even more verbose and add further details. We will also release our code (which is already added to this submission). Please let us know if you have further suggestions and we would be happy to incorporate them.
>
> Thank you for pointing out that a comparison to more advanced RL algorithms would be interesting - this is valuable feedback for us. You mention that PPO and TRPO already have mechanisms for handling variable-length episodes. We politely disagree, as those methods explicitly operate on finite and fixed or infinte time horizons. We would prefer the interpretation that PPO and TPRO somehow fix potential issues with too large step sizes that potentially originate from the "incorrect" gradient scaling, by considering trust regions or by employing clipping. Our attempt, on the other hand, is principled and provides the correct gradient scaling by design, cf. Remark 2.7. In fact, in our experiments it turns out that the "incorrect" scaling factor can be off from the "correct" one by orders of magnitude, see the "effective learning rate" e.g. in Figures 2 - 4.
>
> TRPO aims to maximize the expected reward, while making sure that in each gradient step $ \mathbb{E} [\operatorname{KL}( \widetilde{\pi}(\cdot, s)  | \pi(\cdot, s)) ] \le \delta$ holds, $\widetilde{\pi}$ is the old policy, so the policy is not changed too much. In practice, the constrained optimization is typically conducted with conjugate gradient algorithms and line searches, where a linear approximation of the expected reward and a quadratic approximation of the constrain is employed. This results in
> $$\widetilde{G}^{(k)} := {(H^{-1})}^{(k)} G^{(k)},$$
>
> where $G^{(k)}$ is the policy gradient and $H^{(k)}$ is the Hessian of the estimated KL divergence. One then considers the update
>
> $$\theta^{(k+1)}=\theta^{(k)}+\alpha^{(k)} \sqrt{\frac{2 \delta}{\widetilde{G}^{(k)} \cdot H^{(k)} \widetilde{G}^{(k)}}} \widetilde{G}^{(k)},$$
>
> where $\alpha^{(k)} > 0$ can be found via line search. In the setting of random time horizons, TRPO uses the "incorrect" gradient estimator $G^{(k)}$, however, the scaling and the "line searching" of the step size may cure potentially bad choices. One can readily replace $G^{(k)}$ with our novel formulas that give the "correct" gradient. We anticipate that this should further improve the performance of TRPO, however, due to the limited time in the rebuttal, we need to leave experiments for the next weeks. We will try to incorporate them in the final version of the paper, thus being able to better contextualize the practical implications within contemporary RL research.
>
> PPO makes even further approximations and considers the loss
> $$ \mathcal{L}(\pi) = \mathbb{E}\left[\min \left( \log \pi (a, s) A, \mathrm{clip}\left(\log \pi(a, s), 1-\varepsilon, 1 + \varepsilon \right) A \right) \right], $$
>
> where $A$ is the advantage function and $\varepsilon > 0$ is a hyperparameter. Also here, we can incoporate the findings of our novel gradient estimator, by adding the scaling factor $\mathbb{E}[N + 1]$ to "fix" the "incorrect" gradient estimators, cf. Propositions 2.4 and 2.6.
>
> Finally, we note that in Section 3 we compared the vanilla gradient estimators on purpose in order to properly study the effect of the random time horizons on the (novel) "correct" and (typically used) "incorrect" gradient formulas and in order to exclude confounding effects.
>
> You further ask for a comparison against RL algorithms that have strategies for handling variable time horizons. Could you please let us know which ones you have in mind?

---

### Decision · Program_Chairs · 2025-05-01

**Decision:**

Accept (poster)

**Comment:**

This paper addresses a particular RL setting where the time horizon is a random variable which is common in practice (and especially in LLMs) but poorly studied. In particular, the authors provide corrections to the formulation of the policy gradient that take this specific setting into account. Experiments show that making this correction impacts positively the performance on a set of simple RL benchmarks.

The reviewers appreciated the significance of the problem addressed, which is a more realistic setup for reinforcement learning. They also appreciated the rigor of the methodology.

Several concerns were raised, including the simplicity of the experimental setup, which was considered somewhat toy-ish. The most significant concern relates to a potential lack of novelty. The modification introduced to the policy gradient computation was regarded as relatively straightforward.

This paper may open the path for a more systematic study of this specific setup which relatively common in practice, which seems to be a good contribution to the community.